# Think or Not? Selective Reasoning via Reinforcement Learning for Vision-Language Models

**Jiaqi Wang**[1†]  **Kevin Qinghong Lin**[2†]  **James Cheng**[1]  **Mike Zheng Shou**[2✉]

[1]The Chinese University of Hong Kong    [2]Show Lab, National University of Singapore

## Abstract

Reinforcement Learning (RL) has proven to be an effective post-training strategy for enhancing reasoning in vision–language models (VLMs). Group Relative Policy Optimization (GRPO) is a recent prominent method that encourages models to generate complete reasoning traces before answering, leading to increased token usage and computational cost. Inspired by the human-like thinking process—where people skip reasoning for easy questions but think carefully when needed—we pioneer how to enable VLMs to first decide *when reasoning is necessary*. To realize this, we propose TON, a two-stage training strategy: (*i*) a supervised fine-tuning (SFT) stage with a simple yet effective "**thought dropout**" operation, where reasoning traces are randomly replaced with empty thoughts. This introduces a think-or-not format that serves as a cold start for selective reasoning; (*ii*) a GRPO stage that enables the model to freely explore when to think or not, while maximizing task-aware outcome rewards. Experimental results show that TON can *reduce the completion length by up to* **90**% *compared to vanilla GRPO, without sacrificing performance or even improving it*. Further evaluations across LLM (GSM8K), VLM (CLEVR, Super-CLEVR, GeoQA), and Agentic (AITZ) tasks—covering a range of reasoning difficulties under both 3B and 7B models—consistently reveal that the *model progressively learns to bypass unnecessary reasoning steps as training advances*. These findings shed light on the path toward human-like reasoning patterns in RL approaches. Our code is available at https://github.com/kokolerk/TON.

## 1   Introduction

*"To think or not to think, that is the question."*

Reinforcement learning (RL) has recently emerged as a dominant post-supervised fine-tuning (SFT) strategy in vision-language models (VLMs) [1, 2, 3, 4]. Methods like GRPO [5] have shown promising results in enhancing reasoning capabilities through KL-divergence losses based on rule-driven rewards. However, these approaches often lead to unnecessarily long and redundant reasoning processes due to their reliance on full-length generative trajectories [6, 7, 8]. To address this inefficiency, some works attempt to shorten reasoning chains with rule-based reward penalties [9, 10, 11] during the pre-training phase or introduce external control mechanisms, such as in very recent Qwen3 [12]. Nonetheless, a more natural and scalable solution is to enable the model to *decide when to think*—mirroring how humans modulate cognitive effort in response to task difficulty.

In this work, we begin by presenting empirical evidence that *thinking is not always necessary*. In AITZ [13], we observe that $51\%$ of questions can be answered correctly even when the entire reasoning trace is omitted, resulting in significant savings in thought tokens. This finding underscores

---

† *Equal contribution.*   ✉ *Corresponding authors.*

39th Conference on Neural Information Processing Systems (NeurIPS 2025).

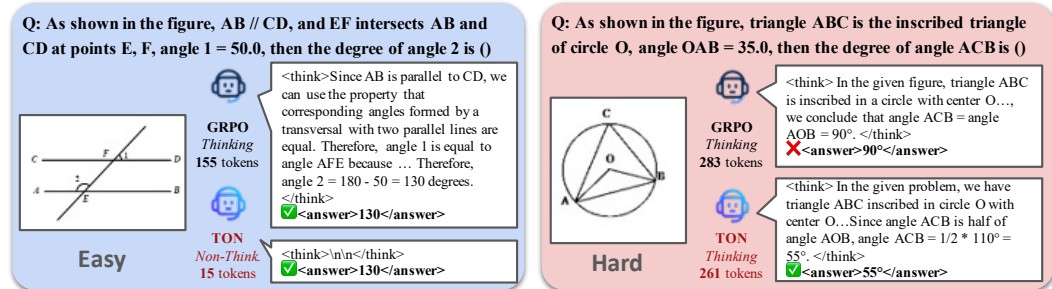

Figure 1: **Illustrating the "think or not think" trade-off. Left**: For simple queries, explicit reasoning is unnecessary—models like GRPO that always "think" incur redundant computation. **Right**: For more complex geometric problems, step-by-step reasoning is essential to arrive at the correct answer. Our proposed TON framework learns to adaptively invoke reasoning only when needed—skipping it for easy cases while engaging in deeper inference for harder tasks.

the potential of selective reasoning strategies to improve efficiency without sacrificing accuracy. Secondly, by exploring a simple prompting strategy — allowing the model to skip reasoning steps for easier queries — we observe that even math-enhanced VLMs struggle to adaptively omit redundant thought generation. Instead, they tend to default to a conservative approach, producing full reasoning traces regardless of task difficulty. This suggests that the ability to "think or not" is not solely determined by reasoning capacity, but should instead be treated as a distinct skill—one that should be explicitly activated through format-following in supervised fine-tuning (SFT) stage.

Motivated by the above observations, we introduce **TON** (*i.e.,* **T**hink-**o**r-**N**ot), a two-stage training framework featuring a simple yet effective "**thought dropout**" approach. This method explicitly replace reasoning traces with minimal "\n\n" delimiter and employs SFT to train the model that reasoning can be skipped—thereby enabling the possibility of bypassing reasoning. A subsequent GRPO stage further refines this selective-reasoning policy via self-exploration, rewarding answers without introducing extra regularization. As illustrated in Figure 1, vanilla GRPO consistently generates reasoning sequences regardless of task difficulty. In contrast, our method, TON, adaptively allocates reasoning based on the complexity of the task. For simple tasks (left), TON can bypass unnecessary reasoning and directly provide the answer, reducing 90% token usage. For more hard problems (right), it still engages in detailed, step-by-step reasoning to arrive at the correct solution. To the best of our knowledge, TON is ***the first work*** *to study "when to think" in VLM.*

Built on top of TON, we using the Qwen-2.5-VL series and conduct extensive evaluations across the LLM benchmark (GSM8K [14]), vision-language tasks—spanning counting (CLEVR [15], SuperCLEVR [16]) as well as mathematical reasoning (GeoQA [17]), and the agent task like mobile agent navigation (AITZ [13])—which collectively cover a spectrum of reasoning levels and diverse task settings. Overall, we find that TON **achieves substantial reductions in completion length without compromising performance**—cutting 87% of tokens on CLEVR and 65% on GeoQA. Notably, on the multi-step navigation task AITZ, TON reduces the average task-level output length from $3.6K$ to $0.9K$ tokens. Moreover, we observe that **omitting reasoning traces can even improve performance**: on GeoQA, TON outperforms the vanilla GRPO baseline by up to 17% in accuracy, demonstrating a "free-lunch" effect where shorter reasoning outperforms or matches longer trajectories. Comprehensive ablation studies further reveal that **the skip-thought ratio increases progressively with reward improvements** during training, suggesting the model learns to selectively bypass unnecessary reasoning steps in an adaptive manner.

## 2    Related Works

**Reinforcement Learning for Vision–Language Models.** Most VLMs start with SFT on large collections of instruction data to acquire broad foundational knowledge [18, 19, 13, 20]. To further improve performance, recent work has adopted a post-training paradigm that leverages human feedback [21, 22, 23]. RL from human feedback (RLHF) fits a reward model on preference annotations and refines the policy via Proximal Policy Optimization (PPO) [24, 25, 22, 26]. Direct Preference Optimization (DPO) [27] streamlines this workflow by recasting policy updates as a binary classification task, aligning model outputs distributions with human preferences without a reward module.

Beyond these methods, Group Relative Policy Optimization (GRPO) [5] blends offline and online learning: it samples groups of thinking process, uses Answer verification (such as Math verifier) as reward feedback, and computes relative advantages within each group. By avoiding a value function, GRPO provide an elegant solution by promoting diverse reasoning paths and improved answer quality. Despite a series of GRPO follow-up works [28, 29, 9], all of these approaches assume that every question demands a full thinking—leading to lengthy decoding. In contrast, our work focuses on "*when to think*" instead of "*how to think*": we introduce a selective reasoning policy that learns to skip unnecessary "think" phases, boosting inference efficiency without sacrificing accuracy.

**Thinking in Language Models.** From early Chain-of-Thought [30, 31, 32] prompting to recent reasoning-intensive reinforcement learning approaches [5, 23, 33, 34], reasoning has emerged as a core dimension in the development of language models. Most existing work emphasizes how to enhance reasoning capabilities, often resulting in increasingly lengthy and complex thought processes [7, 35, 10] while relatively few studies address the efficiency of reasoning. For instance, [36] proposes a long2short strategy to compress decoding length, [37] encourages models to output "I don't know" to terminate unproductive reasoning, and [38] introduces a token-budget-aware reasoning policy. While these approaches offer promising insights into controlling reasoning length, we argue for a more foundational perspective: rather than deciding how to reason once the process has started, *models should first determine whether reasoning is necessary at all*. Simple questions may be answered directly without any explicit reasoning, while complex questions may require maintaining a full reasoning trajectory [8, 6, 9]. In this work, we explore the selective reasoning paradigm within VLMs by introducing a simple yet effective method – **thought-dropout**. We validate its effectiveness on tasks such as Counting, Math, and further extend it to more practical agentic settings.

# 3 Preliminary

**Task Definition.** We formalize the vision-language reasoning environment as a Markov Decision Process (MDP) defined by a tuple $(\mathcal{V}, \mathcal{Q}, \mathcal{S}^*, \pi, r)$, covering a wide range of vision-language tasks. Here, $\mathcal{V}$ denotes the visual context (*e.g.,* an image). $\mathcal{Q}$ is a language-based query or question posed about the visual input. The model, governed by policy $\pi$, takes the input pair $(\mathcal{V}, \mathcal{Q})$ and generates a predicted answer $\mathcal{S}$. The environment provides a scalar reward function $r(\cdot)$ based on the model's response $\mathcal{O}$. A correct prediction, *e.g.,* $\mathcal{O}$ matches the ground truth answer $\mathcal{S}^*$, yields a positive reward, while an incorrect one yields zero. The objective in this environment is to learn an adaptive policy $\pi_\theta$, parameterized by $\theta$, that maximizes the expected reward, enabling the model to reason selectively and efficiently across diverse input settings.

**Reward Function.** The reward function $r(\cdot)$ can be either model-based [24, 26] or rule-based, as recently demonstrated in [5, 23], which is typically categorized into two types: format rewards $r_f$ and outcome rewards $r_o$. While the outcome rewards are usually carefully designed based on different tasks or requests in previous works [5, 1, 2, 9], the format reward $r_f$, is always shared in the same. Given the response $\mathcal{O}$, it should follow the required HTML tag format *<think>*$\mathcal{T}$*<\think><answer>*$\mathcal{S}$*<\answer>*, where $\mathcal{T}$ is the reasoning process (*i.e.,* a thought) and $\mathcal{S}$ is the predicted answer. This formulation requires the model to think before deriving the answer and makes it easy to parse both the reasoning process and the final outcome (*e.g.,* via regular expression).

# 4 TON: Selective Reasoning via Policy Optimization

**Observation.** In practice, humans do not require explicit reasoning for all tasks—many can be completed intuitively. Similarly, models can often produce correct answers to simple questions without explicit thinking. As illustrated in figure 2, the percentages of correct and incorrect samples under different setups with and without the thinking process in inference (see Appendix A for overall performance). We find that $52.1\%$ of answers remained correct without "think," and $14.5\%$ were even correct only without it—implying that *explicit thinking is not always necessary*.

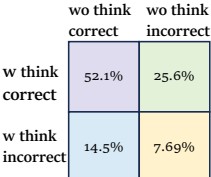

Figure 2: **Accuracy comparison of with *v.s.* without "thinking"** during SFT using Qwen-2.5-VL-3B on the AITZ task.

A straightforward idea is to prompt the model to decide whether to "think" or not (we prompt the model to skip thinking in the simple questions in Sec. 5.4). However, as shown in our experiments (Figure 5d and Appendix G.7), the model still tends to generate the full reasoning process without any no-think try. This suggests that the ability to decide whether to think is not solely governed by reasoning capability, but should instead be treated as a separate skill—one that must be explicitly trained through format-following during the supervised fine-tuning (SFT) stage. These observations motivate us to activate this ability early in the SFT stage and develop TON, which enables selective reasoning by automatically switching between "think" and "non-think" modes.

### 4.1 First SFT stage: Thought Dropout

In the initial stage, the model is typically fine-tuned on "think-answer" formatted data, where the "think" contains high-quality reasoning traces to serve as a cold start. To extend this predefined reasoning ability to selective reasoning, we view "think" vs. "non-think" as part of the output format itself by *dropping* the "think" component during training.

However, it is difficult to determine which samples should be skipped, as different models exhibit varying reasoning capabilities. Therefore, we begin with *random* dropout and allow the model to learn to decide for itself during the second RL stage (Sec.4.2). To this end, we propose "**Thought Dropout**" that randomly injecting empty "thought" segments, requiring only minor code changes:

---

**Algorithm 1** Pseudo-code for `thought_dropout`

```python
def thought_dropout(thought, dropout_prob):
    if random.random() < dropout_prob:
        thought = "\n\n"
    return thought
```

---

This approach injects both the answer format and the skip-thought format as prior knowledge before the second RL stage.

**Where do Thoughts come from?** Given a policy operating in an environment $(\mathcal{V}, \mathcal{Q}, \mathcal{S}^*, \pi, r)$, a key challenge is how to curate high-quality cold-start "thought" data without relying on external models, such as closed-source APIs. A naïve approach is to run multiple inference passes and retain only successful cases based on answer matching—but we find this to be less effective. To address the scarcity of high-quality "thought" data, we instead adopt a **reverse thinking** strategy: leveraging the base model $\pi$ itself to self-generate a rich corpus of thought sequences. Specifically, given the visual context $\mathcal{V}$, textual query $\mathcal{Q}$, and ground-truth answer $\mathcal{S}^*$, we prompt the policy $\pi_\theta$ to deduce the corresponding intermediate thought as follows:

$$\mathcal{T} \leftarrow \pi_\theta(\mathcal{V}, \mathcal{Q}, \mathcal{S}^*) \tag{1}$$

Specially, we generate intermediate thoughts with the following prompts:

> **Prompt for *Reverse Thinking***
>
> Based on the following question and image, generate a thought process to explain how to derive the answer from the inputs.
> Image: {Image} Question: {Question} Answer: {Answer}
> Do not output the answer, only generate the reasoning process. Formulate your outputs using concise language.

In this way, we curate sufficient thought data without relying on external models. These serve as our cold-start training corpus, enabling us to apply the Thought Dropout strategy during SFT to activate the model's ability to bypass thoughts.

### 4.2 Second RL stage: Group Relative Policy Optimization

Although SFT teaches the skip-thought format, it still leaves a central question unresolved: when should thoughts be skipped or retained? Ideally, the model should learn to explore this decision on

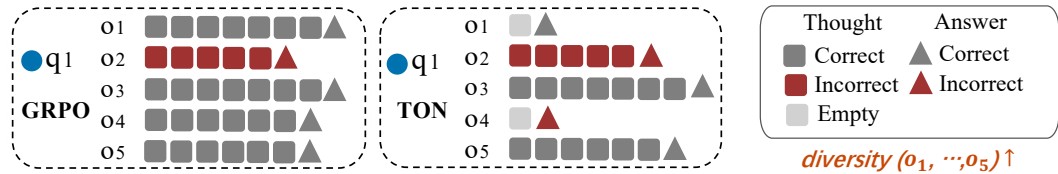

Figure 3: **Illustration of the responses from GRPO and TON.** $q_1$ is the question and $\{o_1, \cdots, o_5\}$ are the generated responses containing thoughts $\mathcal{T}$ (circle) and answers $\mathcal{S}$ (triangle). TON can sample from the empty think $\mathcal{T}_{\backslash n \backslash n}$, thus enhancing the response diversity over the vanilla GRPO.

its own. To this end, we adopt reinforcement learning via GRPO to enhance the model's ability to explore this decision as part of its reasoning process.

Given an image $v \in \mathcal{V}$ and text query $q \in \mathcal{Q}$, GRPO samples $N$ candidate responses with variations $\{o_1, o_2, \ldots, o_N\}$ from the policy $\pi_\theta$ and evaluates each response $o_i$ using a reward function $r(\cdot)$, which measures the quality of the candidate in the context of the given question. To determine the relative quality of these responses, GRPO normalizes the rewards by computing their mean and standard deviation and subsequently derives the advantage as:

$$A_i = \frac{r(o_i) - \text{mean}\{r(o_1), r(o_2), \ldots, r(o_N)\}}{\text{std}\{r(o_1), r(o_2), \ldots, r(o_N)\}} \quad (2)$$

where $A_i$ represents the advantage of the candidate response $o_i$ relative to other sampled responses. GRPO encourages the model to generate responses with higher advantages within the group by updating the policy $\pi_\theta$ using the following objective:

$$\mathcal{J}_{GRPO}(\theta) = \mathbb{E}[\{o_i\}_{i=1}^N \sim \pi_{\theta_{old}}(v, q)] \frac{1}{N} \sum_{i=1}^N \{\min[\alpha_i \cdot A_i, \ \beta_i \cdot A_i] - \beta \mathbb{D}_{KL}[\pi_\theta || \pi_{ref}]\} \quad (3)$$

$$\alpha_i = \frac{\pi_\theta(o_i|v, q)}{\pi_{\theta_{old}}(o_i|v, q)}, \quad \beta_i = \text{clip}\left(\frac{\pi_\theta(o_i|v, q)}{\pi_{\theta_{old}}(o_i|v, q)}, 1 + \epsilon, 1 - \epsilon\right). \quad (4)$$

**How does TON impact GRPO?** As illustrated in Fig. 3, our TON allows the model to choose "empty-think" $\mathcal{T}_{\backslash n \backslash n}$ during the inference step, thus resulting in a significant variation in the distribution between the non-think ($o_i \sim \mathcal{T}_{\backslash n \backslash n}$) and think responses ($o_i \sim \mathcal{T}$) by TON compared to both think ones ($o_i \sim \mathcal{T}$) generated by vanilla GRPO. Unlike previous works like DAPO [28] emphasize on advantage distribution $A_i$ by dynamic sampling in the sparse reward space, our TON shifts the focus to the latent distribution space of responses ($\pi_\theta(o_i|v, q)$), thus enhancing the diversity of the terms $\alpha$ and $\beta$ in Eq. 4.

**How to design Rewards?** To support GRPO training across diverse settings, it is crucial to carefully examine reward design choices. We consider two main types of matching:

*(i) Discrete Matching.* For tasks with deterministic, categorical or numerical outputs—e.g., classification, counting, or math problems—we use a binary value reward $r_d(s, g) = \mathbf{1}(s = g)$: if the predicted answer $s$ matches the ground-truth $g$, we assign $r_d = 1$; otherwise, $r_d = 0$.

*(ii) Continous Matching.* For tasks producing continuous outputs—e.g., spatial coordinates in UI navigation or object grounding—we allow a tolerance region. Given a predicted point $\mathbf{p} = [x, y]$ and a ground-truth box $\mathbf{b} = [x_1, y_1, x_2, y_2]$, we define:

$$r_c(\mathbf{p}, \mathbf{b}) = \begin{cases} 1, & \mathbf{p} \text{ lies inside } \mathbf{b}, \\ 0, & \text{otherwise.} \end{cases}$$

If only a ground-truth point $\mathbf{p}^*$ is available, we use a distance threshold $\theta$:

$$r_c(\mathbf{p}, \mathbf{p}^*) = \begin{cases} 1, & \|\mathbf{p} - \mathbf{p}^*\|_2 \leq \theta, \\ 0, & \text{otherwise.} \end{cases}$$

In practice, we sum the applicable components to form an outcome reward: $r_o = r_d + r_c$. This simple yet flexible scheme can cover classification, numeric reasoning, and grounding. See Appendix B for details on adapting these rewards alongside the format reward to individual downstream tasks.

Table 2: **Performance comparison between TON and vanilla GRPO.** Acc. is the accuracy on the test set. Time is the RL training time. Length is the average competition length at the end of training.

| | Counting (CLEVR) | | | Math (GeoQA) | | | | | |
|---|---|---|---|---|---|---|---|---|---|
| | Acc ($\uparrow$) | Time ($\downarrow$) | Len. ($\downarrow$) | Acc ($\uparrow$) | Time ($\downarrow$) | Len. ($\downarrow$) | Acc ($\uparrow$) | Time ($\downarrow$) | Len. ($\downarrow$) |
| | | Qwen-2.5-VL-3B | | | Qwen-2.5-VL-3B | | | Qwen-2.5-VL-7B | |
| Baseline | 64.0 | - | 306 | 36 | - | 924 | 38 | - | 939 |
| w. GRPO | 93.5 | 1h 44m | 227 | 37 | 2h 50m | 272 | 44 | 3h 4m | 1069 |
| w. TON | 98.5 | 57m | 28 | 51 | 2h 4m | 96 | 61 | 2h 34m | 112 |
| **Gain** | $+5.0$ | $-47m$ | $-87\%$ | $+14$ | $-46m$ | $-65\%$ | $+17$ | $-29m$ | $-90\%$ |

# 5   Experiments

In this section, we conduct experiments on various benchmarks to evaluate our approach. Mainly, we design the experiments to study the following key questions:

**Q1**: Compared to vanilla GRPO, how does TON impact performance and efficiency?

**Q2**: Is there a correlation between TON's skipping behavior and the strength of reasoning ability (*e.g.,* different model sizes or a single model under different iterations)?

**Q3**: Do we really need SFT with thought dropout? Can we rely solely on prompt following if the base model is strong enough?

## 5.1   Benchmarks and Settings

To evaluate the effectiveness and generalization ability of our approach on the below settings:

**Benchmarks.**   We evaluate TON on three vision-language benchmarks, including the general benchmark CLEVR [15] (3D object counting), agent benchmark AITZ [13] (mobile navigation), and the math benchmark GeoQA [17] (middle school math questions) as illustrated in Table 1, spanning a spectrum of reasoning lev-

Table 1: **Summary of benchmark used in our evaluation**.

| Benchmark | Model | OOD | Type | Difficulty | Answer | Thought len. |
|---|---|---|---|---|---|---|
| GSM8K [14] | LLM | | Math | Hard | Number | 939 |
| CLEVR [15] | VLM | | Counting | Easy | Integrate | 586 |
| Super-CLEVR [16] | VLM | ✓ | Counting | Easy | Integrate | – |
| GeoQA [17] | VLM | | Math | Hard | Number | 1652 |
| AITZ [13] | Agent | | GUI | Medium | Action (x) | 283 |
| AITZ (OOD) | Agent | ✓ | GUI | Medium | Action (x) | 283 |

els from simple to complex. To benchmark the model's Out-of-Distribution (OOD) performance, we also evaluate on Super-CLEVR [16] to supplement the CLEVR. AITZ comprises four test domains: we train on the {General} and test on the remaining OOD domains: {Web shopping, Google apps, Install}. We remove the choices in GeoQA and ask the model to generate the answer, enhancing the reasoning complexity. AITZ includes action thought annotations, which we utilize directly, while applying our reverse thinking to generate thoughts for SFT on CLEVR and GeoQA. More benchmark details refer to Appendix E.

**Training details.** We conduct our experiments using Qwen-2.5-VL-Instruct-3B/7B [39] as the base model. All experiments are conducted utilizing 8 NVIDIA H20 GPUs. We train 100 steps for both CLEVR and AITZ, and 300 epochs for GeoQA, given its higher reasoning difficulty level. See setup details in Appendix F. We leverage vLLM [40] to accelerate GRPO training. We add the SFT stage before GRPO as the baseline on the agent task with the same setting as TON because we observe that directly applying GRPO would cause the 0 coordinate reward during the training process, considering its complex output format. For simplicity, we set the dropout probabilities to 50% and examine the impact of different dropout ratios selected from $\{20\%, 50\%, 80\%\}$ in Sec 5.3.

For evaluation, we test all the datasets under the greedy strategy. In CLEVR and GeoQA tasks, where answers are numerical, we measure accuracy by comparing the predicted number to the ground truth. In the AITZ task, where answers are structured as JSON-formatted actions, we report step-level and task-level metrics, including type accuracy (correct action type) and exact accuracy (correct action type and click coordinates) following [18].

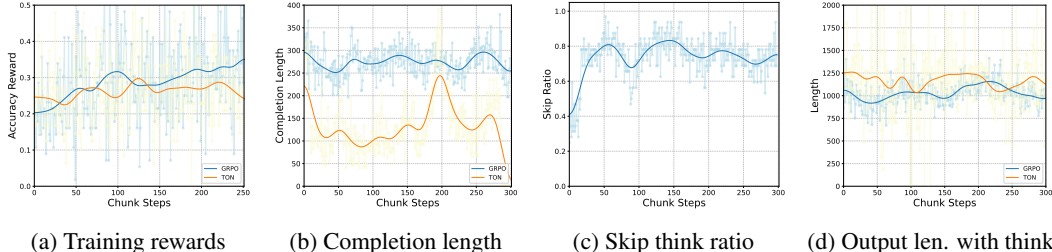

| | (a) Training rewards | (b) Completion length | (c) Skip think ratio | (d) Output len. with think |

Figure 4: **Training metrics comparison between TON and GRPO on GeoQA.** (a) Training rewards, (b) Completion length over training steps, (c) Ratio of non-think outputs to total samples at each step for TON, and (d) Average completion length of think outputs across training.

Table 3: **Out-of-domain (OOD) performance comparison** between our method TON and GRPO on the the AITZ – multi-step mobile navigation. 'Type' is the action type accuracy and 'Exact' requires both the type and value to be correct exactly. 'Avg.' is the average accuracy of all domains. 'Task-level thought's' is the average output lengths on all OOD domains. Step-level accuracy is reported.

| | Think? | IID | | OOD | | | | | | Avg | | Task-level |
| | | General | | Google apps | | Web | | Install | | | | |
| | | type | exact | type | exact | type | exact | type | exact | type | exact | Thought's len. |
| Qwen-2.5-VL-3B | ✓ | 0.01 | 0 | 0.01 | 0 | 0.01 | 0 | 0.01 | 0 | 0.01 | 0 | 2132 |
| w. SFT | ✗ | 0.39 | 0.11 | 0.44 | 0.12 | 0.54 | 0.19 | 0.47 | 0.17 | 0.46 | 0.15 | 742 |
| w. SFT | ✓ | 0.67 | 0.12 | 0.53 | 0.17 | 0.56 | 0.13 | 0.58 | 0.14 | 0.58 | 0.14 | 3572 |
| w. GPRO | ✓ | 0.74 | 0.6 | 0.72 | 0.57 | 0.7 | 0.5 | 0.81 | 0.65 | 0.74 | 0.59 | 3664 |
| w. TON | Ours | 0.74 | 0.6 | 0.74 | 0.56 | 0.72 | 0.5 | 0.78 | 0.64 | 0.75 | 0.59 | 922 |
| *Gain* | | +0.0 | +0.0 | +0.02 | -0.01 | +0.02 | +0.0 | -0.03 | -0.01 | +0.01 | +0.0 | −2742 |
| Qwen-2.5-VL-7B | | 0.28 | 0.14 | 0.26 | 0.1 | 0.33 | 0.13 | 0.39 | 0.16 | 0.31 | 0.13 | 3304 |
| w. GRPO | ✓ | 0.64 | 0.22 | 0.73 | 0.32 | 0.6 | 0.15 | 0.62 | 0.23 | 0.65 | 0.23 | 3272 |
| w. TON | Ours | 0.74 | 0.54 | 0.62 | 0.23 | 0.68 | 0.47 | 0.73 | 0.55 | 0.69 | 0.45 | 908 |
| *Gain* | | +0.1 | +0.32 | -0.11 | -0.09 | +0.08 | +0.32 | +0.09 | +0.32 | +0.04 | +0.22 | −2364 |

## 5.2 Q1: Performance and Efficiency Comparison between TON and GRPO

In Table 2, we present TON on the CLEVR and GeoQA benchmarks under both 3B and 7B settings, with the performance, time consumption, and the average completion length at the RL stage. We find that TON effectively reduces the average of the completion length by up to 90% while achieving comparable even superior performance compared to GRPO with a maximum of 17 Acc. gains. This imply that **skipping unnecessary reasoning can lead to better performance.** The reduction of the completion length decreases the decoding time when generating samples, thus simultaneously shortening the training time. Figure 4a & 4b show the reward and completion length curves where TON remains the rewards on par with vanilla GRPO while the completion length reduces significantly. Appendix G.2 & G.1 shows the entire metrics during training.

**Multi-step Navigation and OOD Testing.** In Table 3, we evaluate TON's performance on AITZ – multi-step mobile navigation, we also assessed its generalization capabilities on OOD test sets using a greedy decoding strategy. Table 3 summarizes the step-level type match accuracy and exact match accuracy for both IID (general) and OOD (Google Apps, web shopping, and install) domains on AITZ, with detailed training visualization in Appendix G.3. Overall, TON demonstrates comparable OOD generalization performance to GRPO, while significantly reducing the task-level output length from $3K$ to $0.9K$ (**70%** **token saving**). This highlights the strong potential of **TON to substantially reduce completion length without compromising performance**. See Appendix G.4 for the OOD performance on other benchmarks.

**Adapt TON framework to the text-only setting.** Furthermore, we extend our study to the LLM domain and present the corresponding experiments and results in Figure 6(b) on the LLM benchmark GSM8K. The findings indicate that TON significantly reduces response length while maintaining high accuracy, demonstrating the generalizability of TON across different modalities.

## 5.3 Q2: Skip Thought Ratio Analysis

Beyond the performance change and completion length reduction achieved by TON, we further investigated the evolution of the skip ratio in 'Thought dropout' during the training step. Figure 4c

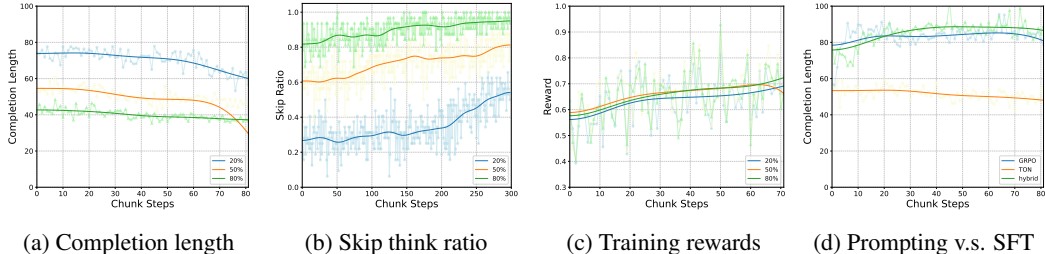

| (a) Completion length | (b) Skip think ratio | (c) Training rewards | (d) Prompting v.s. SFT |

Figure 5: **Further Analysis of TON on the AITZ benchmark.** (a)(b)(c) is the average completion length, skip thought ratios, and the reward under different dropout probabilities. (d) Prompting (hybrid) does not reduce the completion length, while TON using SFT can effectively reduce it.

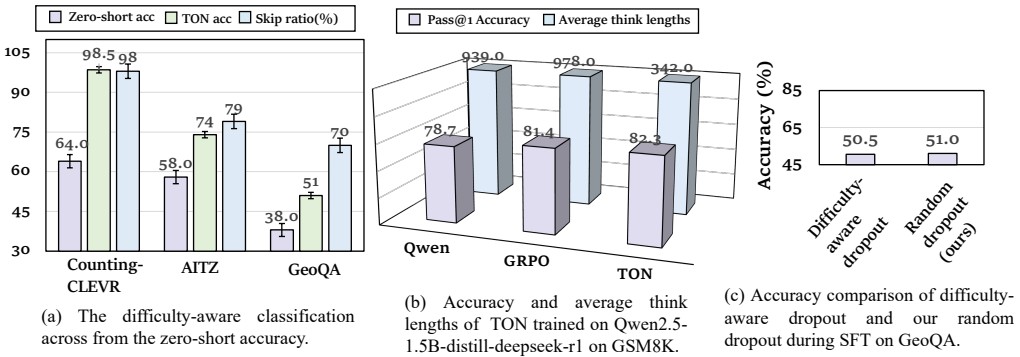

(a) The difficulty-aware classification across from the zero-short accuracy.

(b) Accuracy and average think lengths of TON trained on Qwen2.5-1.5B-distill-deepseek-r1 on GSM8K.

(c) Accuracy comparison of difficulty-aware dropout and our random dropout during SFT on GeoQA.

Figure 6: **Further Analysis of TON.** (a)(c) give the in-depth analysis of the effectiveness of our thought dropout and the generalization across the task difficulty. (b) shows the superiority of TON on the text-only domains.

illustrates the percentage of skip ratio in the generated samples at each step on GeoQA. We observed an increasing trend in the skip ratio during the training process with the increase in training reward. A similar trend is observed across three benchmarks in Figure 21 in the Appendix G.6. This phenomenon suggests that **the model progressively internalizes the reasoning process—learning to skip explicit thoughts while still producing accurate answers.** Moreover, Figure 4d illustrates the length of these outputs generated with 'think'. TON maintain comparable lengths to the vanilla GRPO, indicating that the TON model can **choose not to think but remains diligent when deeper reasoning is necessary.**

**Thought dropout ratio ablation.** We experiment with the impact of different thought dropout ratios of 20%, 50%, and 80% during the SFT stage. Figure 5a & 5b show the completion lengths and the skip ratio during the training process on AITZ. Figure 5c shows a close reward curve of these three variants. Refer more metrics on Appendix G.5. Although the dropout ratios differ, TON consistently exhibits an increasing skip ratio as training progresses. Notably, the 20% setting shows a rapid increase in skip rate, while the higher 80% setting remains relatively stable throughout training. This motivates us to start with a lower dropout probability for further investigation. TON can then be dynamically optimized according to reward signals—decreasing the dropout ratio when performance is high and increasing it when performance drops.

**Deep analysis of the difficulty-aware dropout and random dropout (ours).** As shown in Figure 6(a), we further assess the task difficulty both qualitatively (via intuitive reasoning complexity) and quantitatively (by base VLM zero-shot accuracy), grouped as: 60-100 (easy), 40-60 (medium), and 0-40 (hard). Our findings indicate that our TON maintains a high skip ratio for easy counting queries, allowing it to bypass unnecessary thinking processes, while keeping a low skip ratio for harder, more professional, knowledge-intensive questions to fully utilize its thinking capabilities. Furthermore, we do the difficulty-aware ablation as shown in Figure 6(c), where we drop thoughts for only the easy samples (those answered correctly by the base VLM). To investigate this, we then implement a difficulty-aware dropout strategy and compare its accuracy under our TON training framework where random dropout slightly outperforms difficulty-aware dropout (51.0% vs. 50.5% in

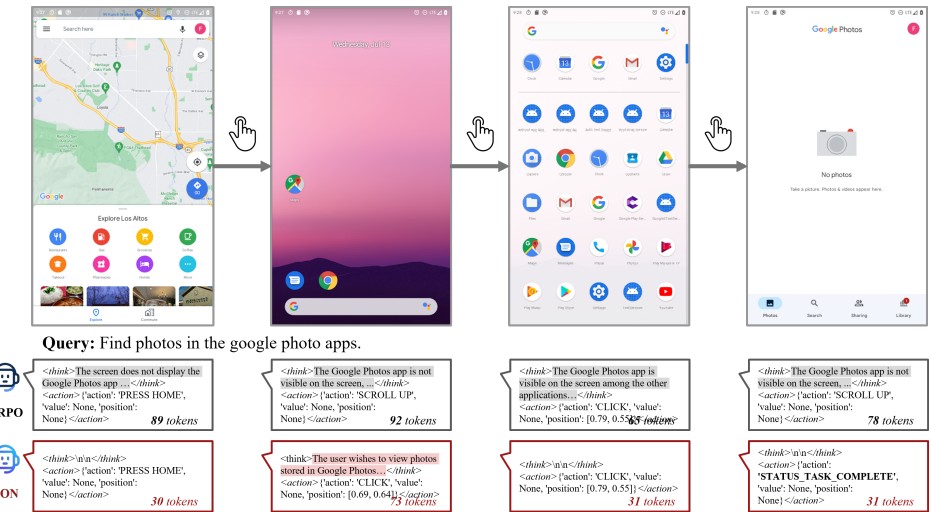

**Query:** Find photos in the google photo apps.

**GRPO**

*<think>* The screen does not display the Google Photos app ... *</think>*
*<action>* {'action': 'PRESS HOME', 'value': None, 'position': None} *</action>*          **89 tokens**

*<think>* The Google Photos app is not visible on the screen, ... *</think>*
*<action>* {'action': 'SCROLL UP', 'value': None, 'position': None} *</action>*          **92 tokens**

*<think>* The Google Photos app is visible on the screen among the other applications... *</think>*
*<action>* {'action': 'CLICK', 'value': None, 'position': [0.79, 0.55]} *</action>* **65 tokens**

*<think>* The Google Photos app is not visible on the screen, ... *</think>*
*<action>* {'action': 'SCROLL UP', 'value': None, 'position': None} *</action>*          **78 tokens**

**TON**

*<think>*\n\n*</think>*
*<action>* {'action': 'PRESS HOME', 'value': None, 'position': None} *</action>*          **30 tokens**

*<think>* The user wishes to view photos stored in Google Photos... *</think>*
*<action>* {'action': 'CLICK', 'value': None, 'position': [0.69, 0.64]} *</action>* **73 tokens**

*<think>*\n\n*</think>*
*<action>* {'action': 'CLICK', 'value': None, 'position': [0.79, 0.55]} *</action>*          *31 tokens*

*<think>*\n\n*</think>*
*<action>* {'action': **'STATUS_TASK_COMPLETE'**, 'value': None, 'position': None} *</action>*          *31 tokens*

Figure 7: **Comparison between GRPO and TON on Agent setting AITZ [13]**. TON adaptively skips unnecessary think steps during multi-step mobile navigation, achieving greater decoding efficiency compared to GRPO while maintaining task accuracy (saving 60% tokens in this case).

TON Accuracy). The results suggest that hand-crafted heuristics for task difficulty may introduce noise or unintended bias, potentially interfering with the learning process while our random dropout offers a simpler, unbiased alternative that generalizes well across tasks.

## 5.4 Q3: Emprical Verfication of SFT Significance in TON

In addition to incorporating the skip-think format during the SFT stage as in TON, we explored a simpler alternative: modifying the prompt to encourage the model to automatically omit reasoning steps, enabling direct GRPO training without the need for a separate SFT stage. The hybrid-thought prompt is defined as follows:

> **Prompt for *Hybrid Thinking***
>
> A conversation between User and Assistant. The user asks a question, and the Assistant solves it. The assistant evaluates whether the question is simple enough to answer directly. If simple, the output is formatted as <think>\n\n\think><answer>answer here\answer>. If the question is difficult, the assistant needs to first think then answering the question. The output is formatted as <think> reasoning process here \think><answer> answer here \answer>. The assistant is encouraged to use <think>\n\n\think> while maintaining accuracy.

Figure 5d shows the completion length of GRPO using the hybrid prompt, vanilla GRPO (with a full-think prompt), and TON throughout the training process on AITZ. Appendix G.7 presents similar trends across three benchmarks, revealing only minor differences in completion length between the hybrid prompt and vanilla GRPO. Moreover, we observe only 2 'skip' cases in GeoQA and none in AITZ among all samples generated by GRPO during both training and inference. We attribute this to the model's tendency to play it safe by generating long and detailed reasoning, consistent with its ingrained behavioral patterns learned during pre-training or SFT. Since the model does not produce skip-thought outputs, applying additional reward to these outputs has no effect, resulting in a zero contribution throughout training. These findings highlight the necessity of our SFT stage with thought dropout (Sec. 4.1) to establish the desired format-following behavior.

## 5.5 Qualitative Examples

Figure 7 compares GRPO and TON on the AITZ benchmark for multi-step mobile navigation. While GRPO generates verbose reasoning at every step, TON adaptively skips unnecessary thinking, reducing token usage by 60% without sacrificing task accuracy. This demonstrates TON's efficiency in handling real-world, *long-horizon procedural agent tasks*. Table 4 further illustrates TON's ability

Table 4: **Illustration between Thinking and Non-Thinking modes from CLEVR** [15]. TON demonstrates selective activation of reasoning—engaging thought only when needed—whereas GRPO generates reasoning traces for both cases indiscriminately. The full outputs are in Table 11.

| Counting examples from CLEVR [15] | | |
| --- | --- | --- |
| |  |  |
| Question (GT) | How many items are there in the image? <answer>3</answer> | How many items are there in the image? <answer>8</answer> |
| GRPO's outputs | <think> The image shows three objects ... 
 1. Green Cylinder: Transparent (metallic) 2. Teal Sphere: Transparent (metallic) 
 </think> 
 <answer>3</answer> | <think> - I count six distinct objects in the picture: a large green matte cylinder, a blue matte cylinder, ... 
 - I also note the size and color variety of each object as described by the visual cues. 
 </think> 
 <answer>10</answer> |
| TON's outputs | <think>\n\n</think> 
 <answer>3</answer> | <think> Okay, let's check this out. The user provided a image with .... 
 First, I need to count ... 
 1. "A small purple rubber sphere..."... 3. "A small gray rubber cube..." Third. 
 ... 
 Wait, that's eight distinct entries... </think> 
 <answer> 8 </answer> |

to selectively activate reasoning. Unlike GRPO, which consistently generates detailed thought traces, TON omits reasoning for simple questions that can be answered at a glance, while producing accurate and focused reasoning for complex scenarios involving visual occlusion.

## 6   Conclusion

We present TON, a simple yet effective two-stage training framework that enables vision-language models to learn *when to reason*—introducing selective reasoning as a controllable and trainable behavior. By combining thought dropout during supervised fine-tuning with reward-guided refinement via GRPO, TON significantly reduces completion length (up to **90**%) without sacrificing—and in some cases improving—performance across diverse reasoning tasks. Our findings challenge the assumption that full reasoning traces are always beneficial and pave the way for more efficient, human-like reasoning strategies in both multimodal intelligence and reinforcement learning.

## 7   Acknowledgements

This research is supported by the National Research Foundation, Singapore under its AI Singapore Programme (AISG Award No: AISG3-RP-2022-030).

We sincerely thank Dongchi Huang for his invaluable guidance on the code and for providing essential computational resources. We also appreciate Binghui Xie's insightful discussion on topic selection and idea suggestions. Additionally, we are grateful to Qiguang Chen and Yuxuan Wan for their thoughtful and constructive feedback on this paper. Finally, we extend our gratitude to Xiaojun Guo and Qixun Wang for their valuable advice on visual reasoning and the GRPO series methods.

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

# Appendix

## Contents

# A  Motivation Experiments

Table 5 presents the performance of the agent VQA with and without the think source during the SFT stage, as well as with and without the think format in the GRPO reward function. Using the think source results in higher performance but longer output lengths, while excluding it leads to lower performance with shorter outputs.

Table 5: Qwen2.5-VL-3B on the agent dataset (Android-in-the-zoo) with/ without think before the answer by SFT and vanilla GRPO. acc is the test accuracy and len is the output length of step-level.

|          | SFT  |     | GRPO |     |
|----------|------|-----|------|-----|
|          | acc  | len | acc  | len |
| w think  | 0.67 | 181 | 0.74 | 88  |
| wo think | 0.39 | 29  | 0.58 | 29  |

# B  Rewards for Downstream Tasks

General VQA: $r = r_f + r_d$, where $r_f$ is the format reward and $r_f = 1$ if the response follows the think answer format, otherwise 0, $r_d$ is the concrete reward that $r_d = 1$ if the predicted answer is equal to the ground truth number.

Agent VQA: $r = r_f + r_d + r_c$, where $r_f$ is the format reward and $r_f = 1$ if the response follows the think answer format, otherwise 0, $r_d$ is the concrete reward that $r_d = 1$ if the predicted action type is equal to the ground truth action type,*e.g.,*`click, press_home`, $r_c$ is the continues reward for the predicted coordinates when the action type is click. In this paper, we use the normalized coordinates ranging from 0-1 and set $\theta = 0.14$ following [19].

Math VQA: $r = r_f + r_d$, where $r_f$ is the format reward and $r_f = 1$ if the response follows the think answer format, otherwise 0, $r_d$ is the concrete reward that $r_d = 1$ if the predicted answer is equal to the ground truth number.

# C  Limitations

Due to computational resources, our current work focuses on smaller-sized visual-language models like 3B and 7B, the proposed method has not been evaluated on even larger models (*e.g.,*235B). We implement TON on the open-domain VLMs; however, without access to the source code of proprietary VLMs like GPT-4o, the proposed method has not been implemented on them.

# D  Broader Impact

In this paper, we propose a simple yet effective method TON, to cooperate SFT and RL stages by thought dropout. We improve the vanilla GRPO's performance by sampling minor code changes to teach the model to reason during the RL exploration stage selectively. This enables a deeper understanding of RL in VLMs, inspiring flexible injection of prior knowledge into the SFT stage instead of manually creating rule-based rewards. For social impact, this work has a certain impact on the RL research in the VLM and LLM.

# E  Dataset

**General VQA.** The CLEVR dataset [15] is designed to generate complex multi-step questions based on synthetic images, assessing a model's true reasoning ability. It is a diagnostic dataset that includes 100,000 rendered images and approximately one million automatically generated questions, of which 853,000 are unique. The dataset features challenging questions involving counting, comparison, logical reasoning, and memory storage, while the images depict simple 3D shapes. In contrast to the original CLEVR dataset, Super-CLEVR [16] introduces more complex visual components and

offers better control over the factors contributing to domain shift. For our experiments, we select a subset of 1,000 datasets that contain only counting problems for training. We evaluate the model's performance on test sets by selecting 200 samples from CLEVR that were not seen in the training set, as well as 200 counting problems from the out-of-distribution Super-CLEVR dataset.

**Math VQA.** GeoQA [17] is a large-scale geometric question answering dataset that contains 4,998 geometric problems collected from real math exams in Chinese middle school. Each problem is accompanied by annotated programs illustrating the solution process. While this dataset features multiple-choice questions, we increase the difficulty in this paper by removing the answer choices and requiring the model to generate the answers directly. We select a subset of 1k problems that involve computing angles and side lengths for training and test the model on this training set.

**GUI Agent.** AITZ [13] is a dataset designed for the graph user interface (GUI) navigation task derived from the large-scale mobile benchmark Android-in-the-wild (AITW [41]). It features a unique annotation called chain-of-action thought (CoAT), establishing a connection between perception—specifically, the understanding of screen layouts and UI elements—and cognition, which involves action decision-making. The AITZ dataset includes 2,504 operational trajectories that encompass 18.6K real-world intentions. Additionally, it is categorized into five subsets based on application domains: General, Install, GoogleApps, Single, and WebShopping. We train the model using the General domain with a dataset of randomly selected 1k examples and evaluate its performance on the corresponding test sets, as well as on other out-of-distribution domains.

# F   Setup

We use Llamafactory [42] for the SFT stage with full parameters, and the training time is no longer than 15 minutes for both Qwen2.5-VL-3B/7B models. We set $\theta = 0.14$ following [19]. We use vLLM [40] and the zero1_no_optimizer GRPO settings to optimize further:

Table 6: Training Parameters for the first SFT of TON

| Parameter | Value |
|---|---|
| cutoff_len | 2048 |
| per_device_train_batch_size | 8 |
| gradient_accumulation_steps | 1 |
| learning_rate | 1.0e-5 |
| lr_scheduler_type | cosine |
| warmup_ratio | 0.1 |
| epoch | 2 |

Table 7: Training Parameters for the second GRPO stage of TON in general/agent

| Parameter | Value |
|---|---|
| max_prompt_length | 4096 |
| max_completion_length | 2048 |
| per_device_train_batch_size | 1 |
| gradient_accumulation_steps | 2 |
| learning_rate | 1e-6 |
| lr_scheduler_type | constant |
| bf16 | true |
| $\beta$ | 0.04 |
| gradient_checkpointing | true |
| attn_implementation | flash_attention_2 |
| min_pixels | 3136 |
| max_pixels | 501760 |
| temperature | 1.0 |
| num_generations | 8 |
| step | 100 |

Table 8: Training Parameters for the second GRPO stage of TON in math

| Parameter | Value |
|---|---|
| max_prompt_length | 4096 |
| max_completion_length | 2048 |
| per_device_train_batch_size | 1 |
| gradient_accumulation_steps | 2 |
| learning_rate | 1e-6 |
| lr_scheduler_type | constant |
| bf16 | true |
| $\beta$ | 0.04 |
| attn_implementation | flash_attention_2 |
| min_pixels | 3136 |
| max_pixels | 501760 |
| temperature | 1.0 |
| num_generations | 4 |
| step | 300 |

# G  Experiments

## G.1  TON on Math–GeoQA

Figure 8 & 9 illustrate the progression of various variables throughout the training process.

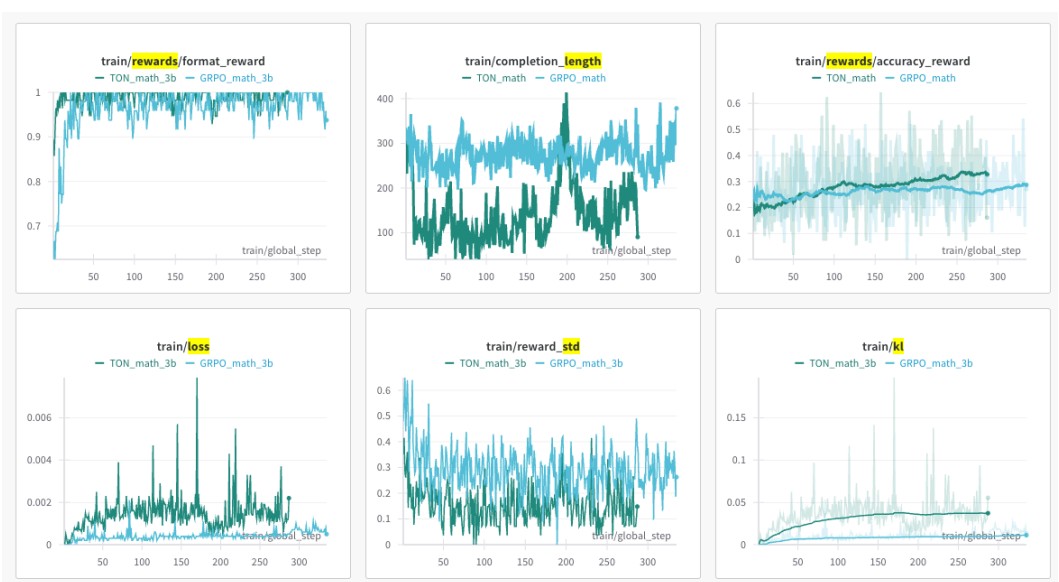

Figure 8: TON and GRPO visualization during the training process on Qwen2.5-VL-3B on GeoQA.

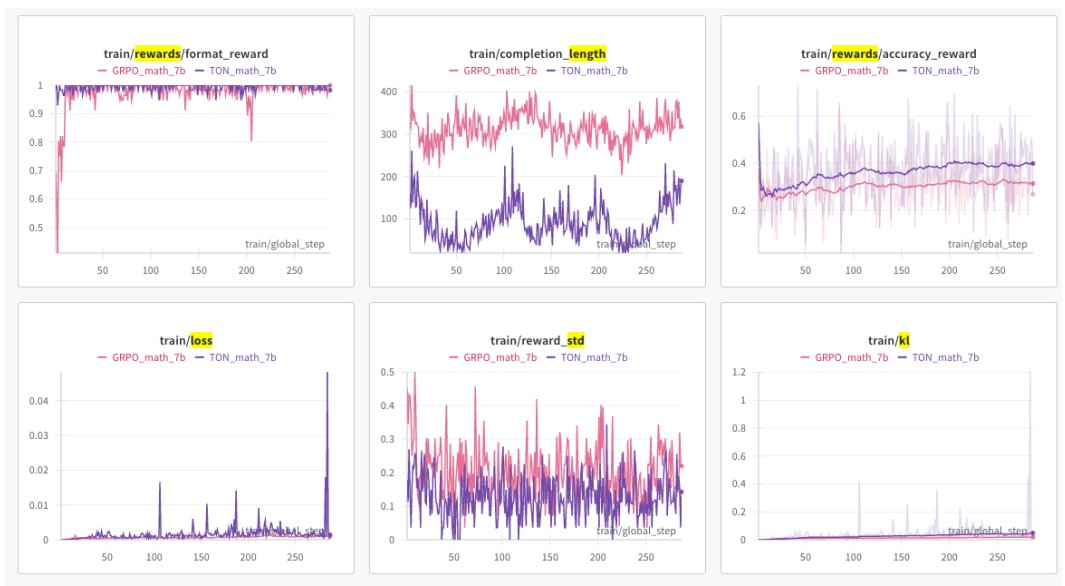

Figure 9: TON and GRPO visualization during the training process on Qwen2.5-VL-7B on GeoQA.

## G.2 TON on Counting–CLEVR

Figure 10 illustrates the progression of various variables throughout the training process.

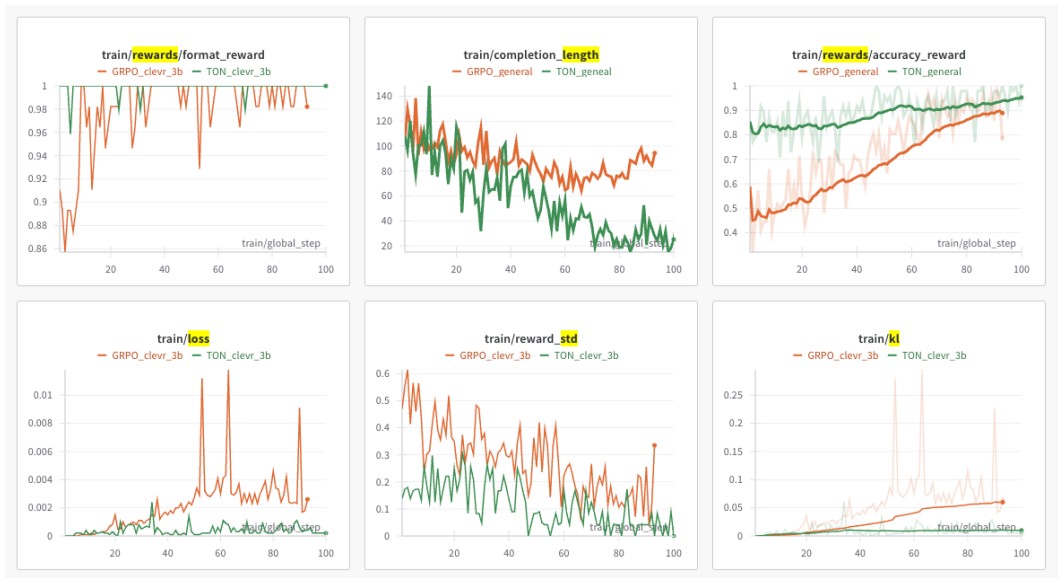

Figure 10: TON and GRPO visualization during the training process on Qwen2.5-VL-3B on CLEVR.

## G.3 TON on Mobile Agent–AITZ

Figure 11 & 12 illustrate the progression of various variables throughout the training process.

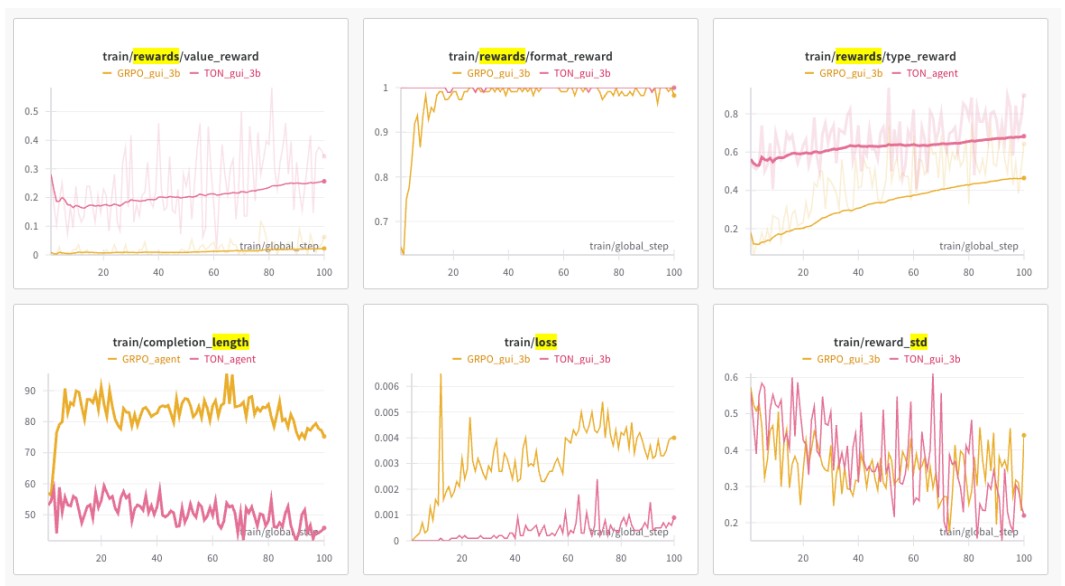

Figure 11: TON and GRPO visualization during the training process on Qwen2.5-VL-3B on AITZ.

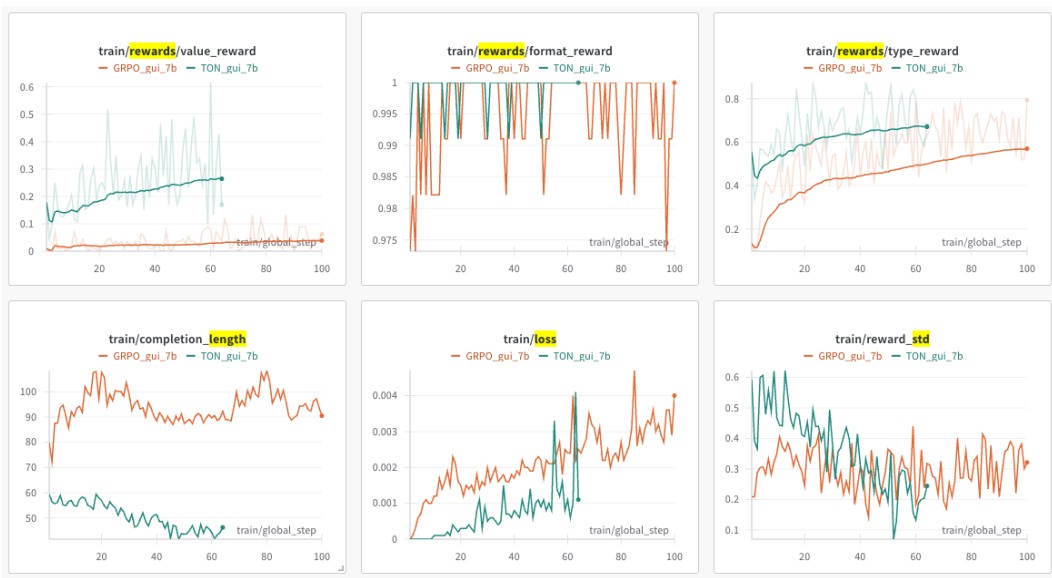

Figure 12: TON and GRPO visualization during the training process on Qwen2.5-VL-7B on AITZ.

### G.4 OOD Performance of TON on CLEVR

Table 9 compares the IID and OOD performance of TON and vanilla GRPO. TON demonstrates superior performance in both IID and, particularly, OOD scenarios under easy reasoning tasks, helping to avoid overfitting to the training set of vanilla GRPO.

Table 9: Qwen2.5-VL-3B on the IID domain CLEVR and OOD domain Super-CLEVR.

| | think | CLEVR acc | Super-CLEVR acc |
|---|---|---|---|
| base | | 64 | 57.3 |
| SFT | ✓ | 88.5 | 13.17 |
| GRPO | ✓ | 93.5 | 51.9 |
| TON | ours | 98.5 | 62.79 |

## G.5 Different Thought Dropout Probabilities

Figure 13 illustrates the progression of various variables throughout the training process under different dropout probabilities.

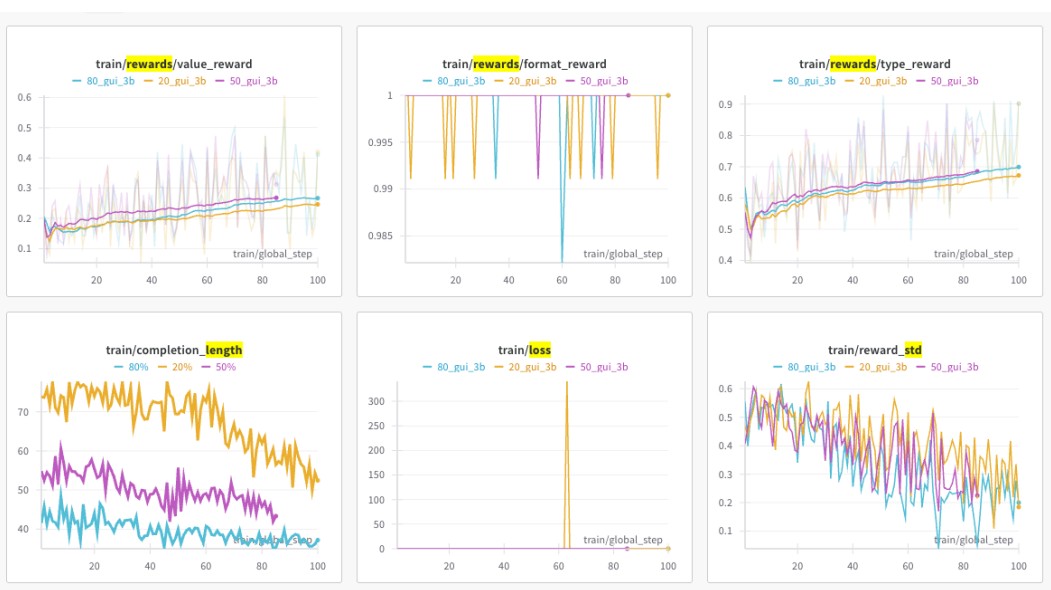

Figure 13: GRPO visualization during the training process on Qwen2.5-VL-3B on AITZ under dropout probabilities 20%, 50%, 80%.

## G.6 Skip-thought Ratio on Different benchmarks

Figure 21 illustrates the skip-thought ratios under TON throughout the training process under different VQA benchmarks.

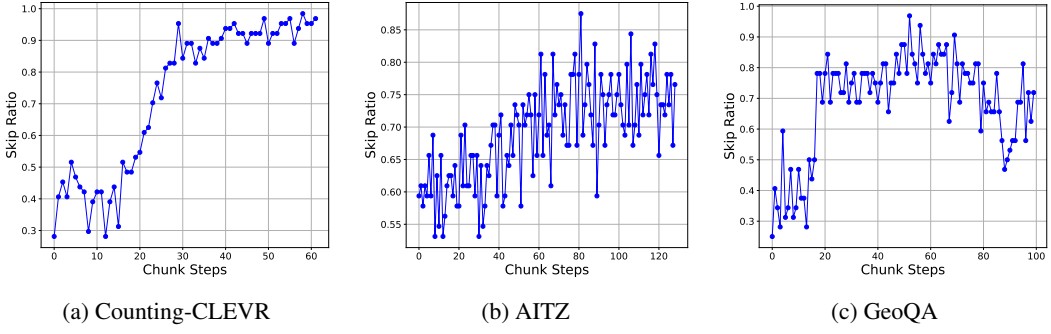

(a) Counting-CLEVR      (b) AITZ      (c) GeoQA

Figure 14: **Skip Ratio of the output thinking** during our TON training on three benchmarks.

## G.7 Prompt v.s. SFT on different benchmarks

Figure 15 & 16 & 17 illustrate the progression of various variables throughout the training process between injecting the skip-thought during the prompt and the SFT stage.

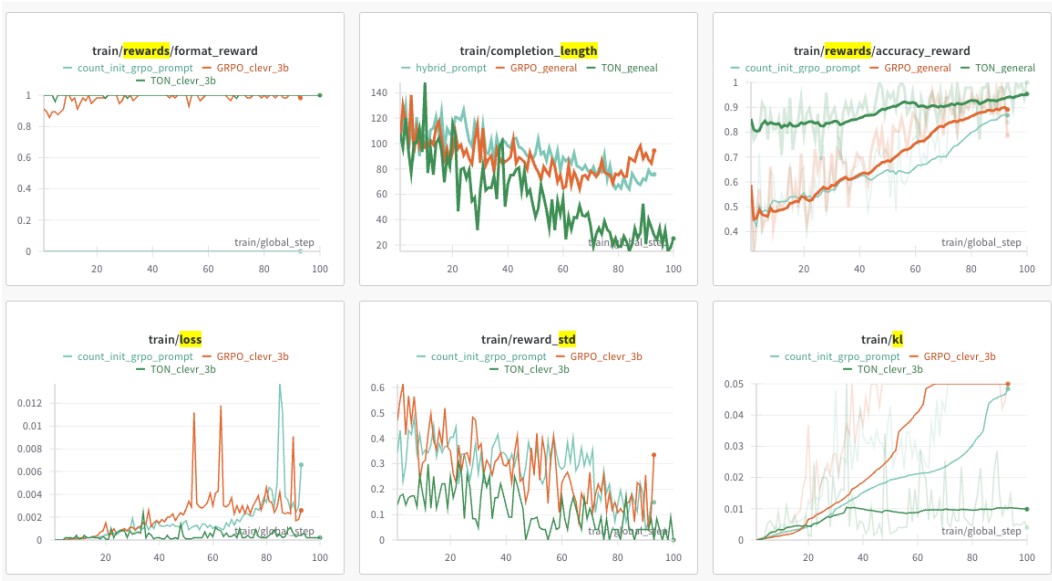

Figure 15: hybrid prompt v.s. SFT visualization during the training process on Qwen2.5-VL-3B on clevr.

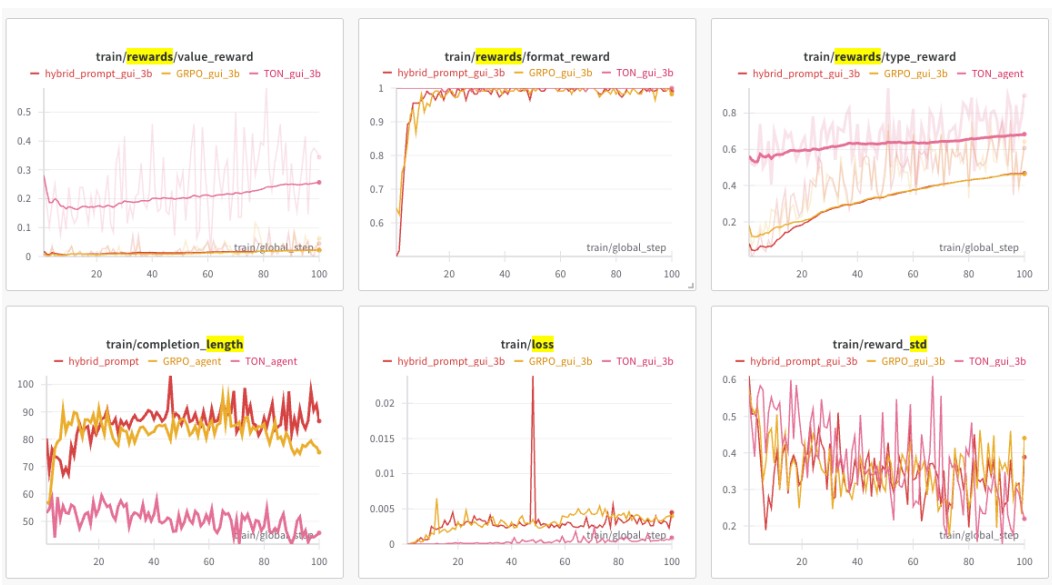

Figure 16: hybrid prompt v.s. SFT visualization during the training process on Qwen2.5-VL-3B on AITZ.

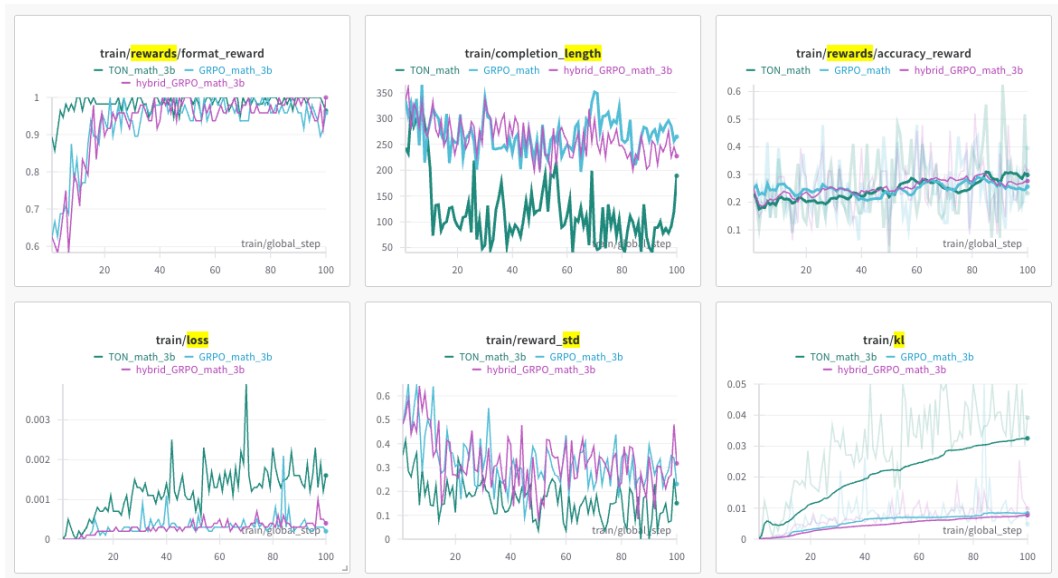

Figure 17: hybrid prompt v.s. SFT visualization during the training process on Qwen2.5-VL-3B on GeoQA.

## G.8 Visualization Examples

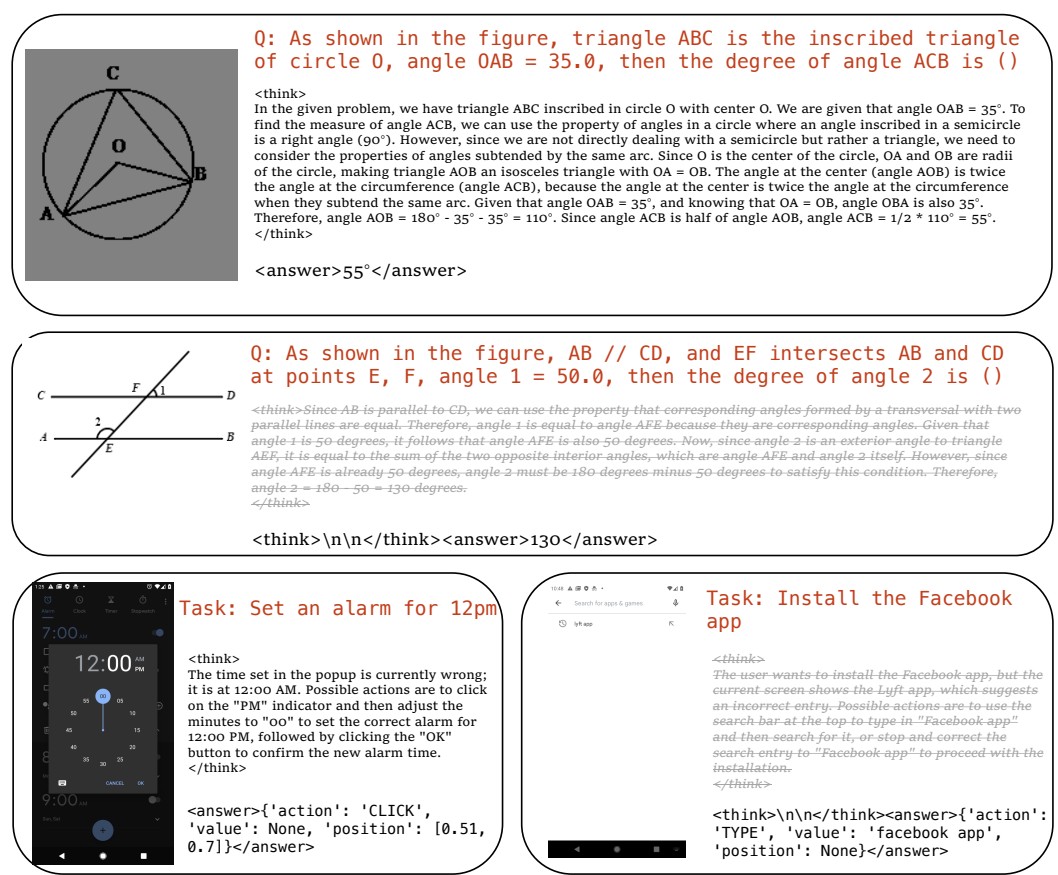

Figure 18: Examples of TON on math VQA and GUI agent VQA settings.

**Counting examples from CLEVR [15]**

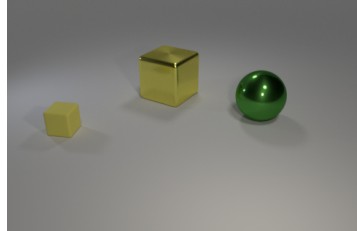 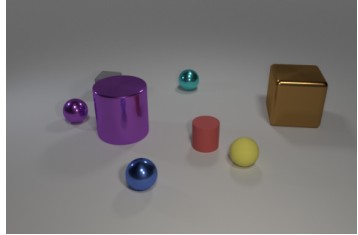

| | | |
|---|---|---|
| Question | How many items are there in the image? | How many items are there in the image? |
| GRPO's outputs | <think>The image shows three objects: a large green cylinder from 1919, a large teal sphere, and a large golden sphere. Both the teal and golden spheres appear to be in a translucent state, indicating that they are metallic. The green cylinder is plain and opaque, without any reflective properties. 1. Green Cylinder: Transparent (metallic) 2. Teal Sphere: Transparent (metallic) 3. Golden Sphere: Transparent (metallic) The question seems to ask for the count of items, and considering that all three objects are distinguishable from each other as separate items in the image, the count is three. </think> <answer>3</answer> | <think> - I count six distinct objects in the picture: a large green matte cylinder, a blue matte cylinder, a large green reflective sphere, a large blue matte cube, a small blue matte cube, a light blue rubber object, a shiny silver cube, a large reflective blue object, a silver or dark green reflective sphere, a shiny purple metallic cylinder, and a blue matte sphere. - I observe the reflective surface properties of the objects around the central large green sphere. - I also note the size and color variety of each object as described by the visual cues. </think> <answer>10</answer> |
| TON's outputs | <think>\n\n</think> <answer>3</answer> | <think>Okay, let's check this out. The user provided a image with several objects and their properties. Each entry starts with "A" followed by attributes like size, color, material, shape, rotation, 3D coordinates, and pixel coordinates. First, I need to count how many items there are. Let me list each entry one by one. 1. "A small purple rubber sphere..." That's the first item. 2. "A large yellow rubber cube..." Second item. 3. "A small gray rubber cube..." Third. 4. "A large green rubber cylinder..." Fourth. 5. "A small blue metal cube..." Fifth. 6. "A small green metal sphere..." Sixth. 7. "A large brown metal cube..." Seventh. 8. "A small green rubber cylinder..." Eighth. Wait, that's eight distinct entries. Each one has unique combinations of attributes. I can't see any duplicates. Each has different values for color, size, material, rotation, or coordinates, so they're separate. So yes, there are eight items in total. </think> <answer> 8 </answer> |

Table 10: Counting example from CLEVR [15]. Although the question is simple, the two conditioned images differ significantly in difficulty: the left image is clearly easier, while the right involves greater complexity due to object occlusion. TON demonstrates selective activation of reasoning—engaging thought only when needed—whereas GRPO generates reasoning traces for both cases indiscriminately.

### G.9 Prompt for AITZ task

**AITZ System Prompt**

You are an assistant trained to navigate the mobile phone. Given a task instruction, a screen observation, and an action history sequence, output the next action and wait for the next observation.
Here is the action space:

- 1. 'CLICK': Click on an element, value is not applicable and the position [x,y] is required.
- 2. 'TYPE': Type a string into an element, value is a string to type and the position is not applicable.
- 3. 'SCROLL UP': Scroll up for the screen.
- 4. 'SCROLL DOWN': Scroll down for the screen.
- 5. 'SCROLL LEFT': Scroll left for the screen.
- 6. 'SCROLL RIGHT': Scroll right for the screen.
- 7. 'PRESS BACK': Press for returning to the previous step, value and position are not applicable.
- 8. 'PRESS HOME': Press for returning to the home screen, value and position are not applicable.
- 9. 'PRESS ENTER': Press for submitting the input content, value and position are not applicable.
- 10. 'STATUS TASK COMPLETE': Indicate the task is completed, value and position are not applicable.

Format the action as a dictionary with the following keys: {'action': 'ACTION TYPE', 'value': 'element', 'position': [x,y]}
If value or position is not applicable, set it as 'None'. Position represents the relative coordinates on the screenshot and should be scaled to a range of 0-1.

**Think Prompt**

Please first thinks about the reasoning process in the mind and then provides the user with the action. The reasoning process and answer are enclosed within <think> </think> and <action> </action> tags, respectively, i.e., <think> reasoning process here </think><action> action here </action>

**Overall Prompt**

{AITZ System Prompt}
{Think Prompt}
Task: {Task}
<image>
Action history: {Action history}
**Next action:**

# H  Supplementary

## H.1  Comprehensive Comparison of Length, Rewards, and Skip Ratio Across Three Benchmarks

We present a comprehensive comparison of length, rewards, and skip ratio across three benchmarks. The results reveal a consistent trend: TON reduces completion length and increases the skip ratio as rewards increase during training.

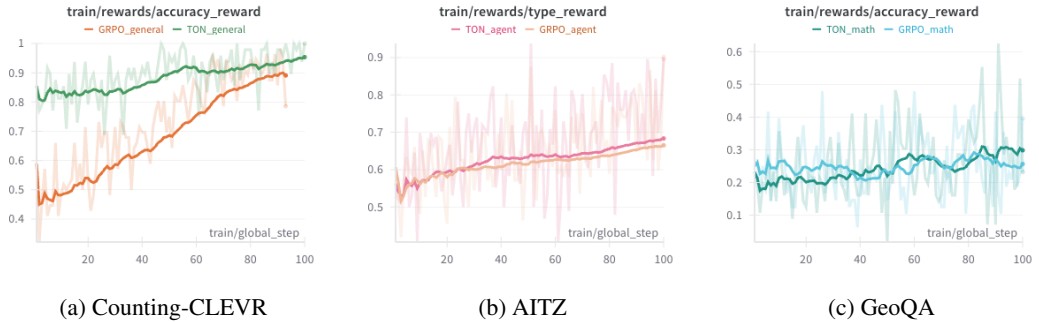

(a) Counting-CLEVR          (b) AITZ          (c) GeoQA

Figure 19: **Rewards of the output** during our TON training on three benchmarks.

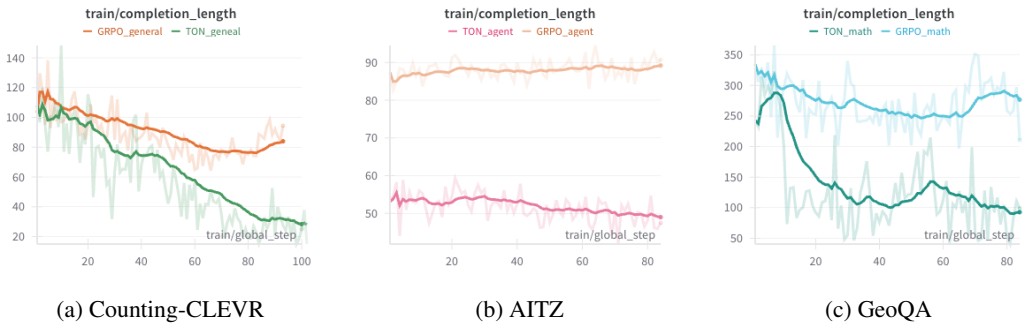

(a) Counting-CLEVR          (b) AITZ          (c) GeoQA

Figure 20: **Completion length of the output** during our TON training on three benchmarks.

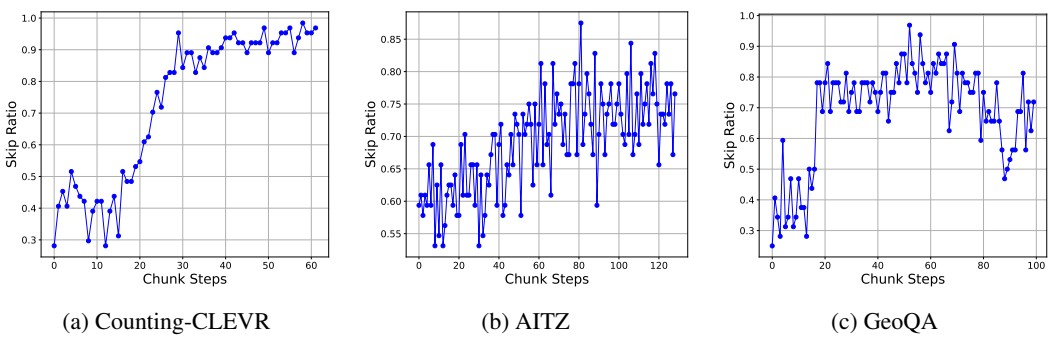

(a) Counting-CLEVR          (b) AITZ          (c) GeoQA

Figure 21: **Skip ratio of the output thinking** during our TON training on three benchmarks.

## H.2  Reward for length rather than SFT

We give the reward $r_l = 1$ for the model if it outputs <think>\n\n<\think>, and otherwise 0. We observe that the length reward remains at 0 during the first 100 steps. The visualization of the entire training process is shown below, highlighting our proposed thought dropout in the SFT stage.

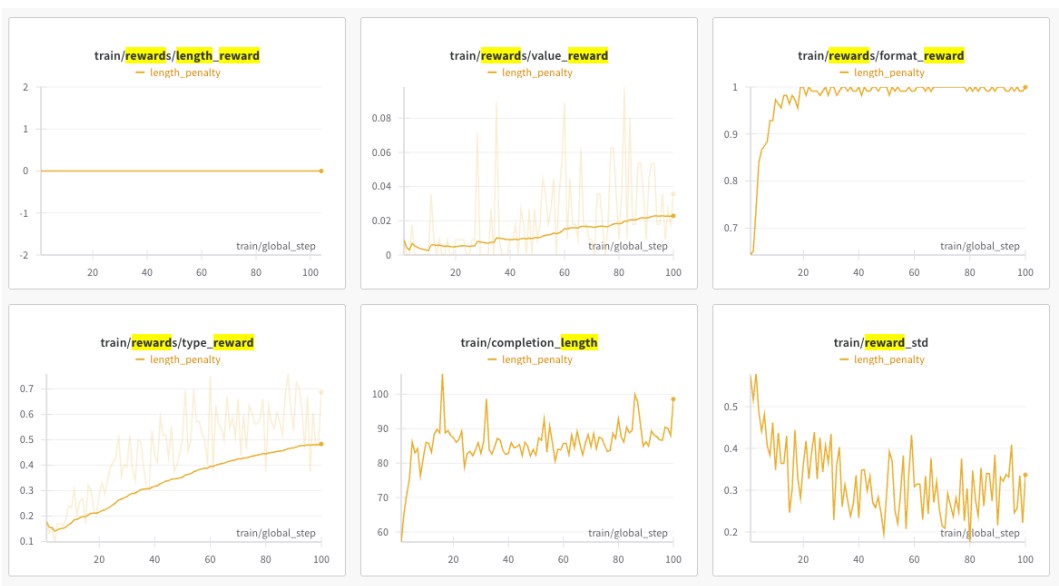

Figure 22: Length penalty rewards and completion length on AITZ datasets on Qwen2.5-VL-3B.

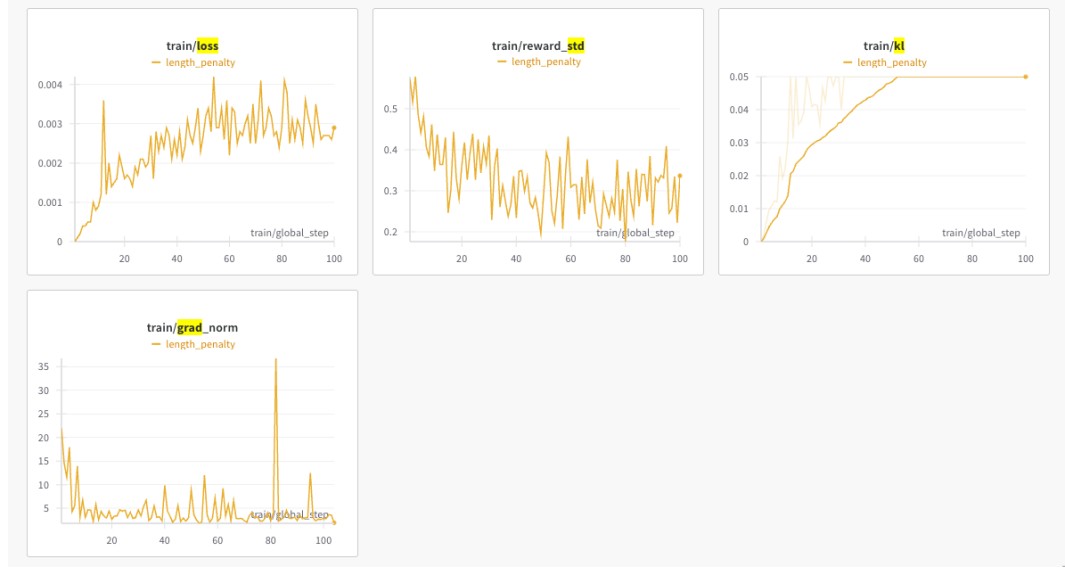

Figure 23: Length penalty rewards and completion length on AITZ datasets on Qwen2.5-VL-3B.

## H.3 More cases

We give more cases to show the effectiveness and efficiency of our proposed TON.

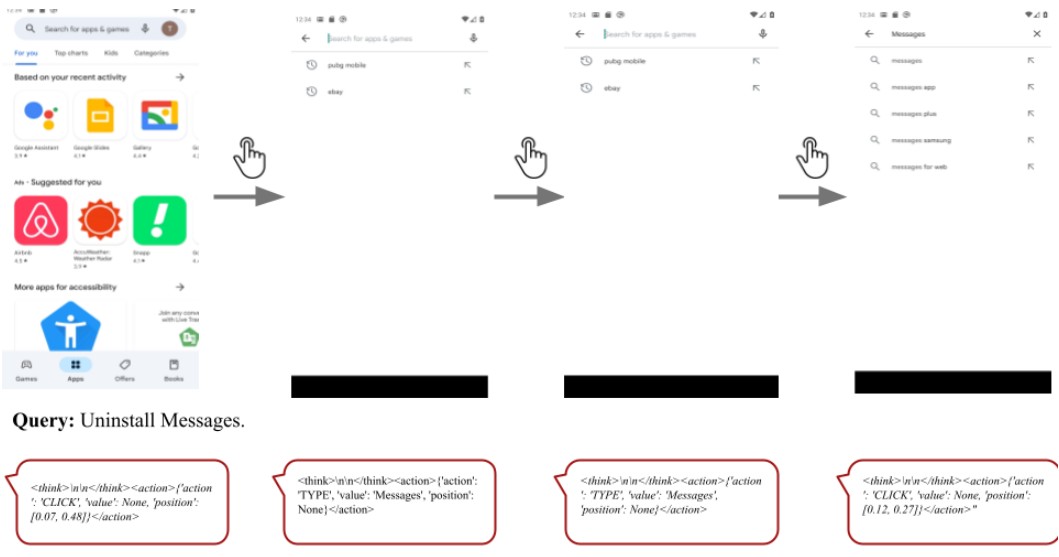

Query: Uninstall Messages.

*<think>\n\n</think><action>{'action': 'CLICK', 'value': None, 'position': [0.07, 0.48]}</action>*

<think>\n\n</think><action>{'action': 'TYPE', 'value': 'Messages', 'position': None}</action>

*<think>\n\n</think><action>{'action': 'TYPE', 'value': 'Messages', 'position': None}</action>*

*<think>\n\n</think><action>{'action': 'CLICK', 'value': None, 'position': [0.12, 0.27]}</action>"*

Figure 24: TON applied to the AITZ out-of-distribution domain: the task is to uninstall the messaging apps. TON performs well without extensive reasoning.

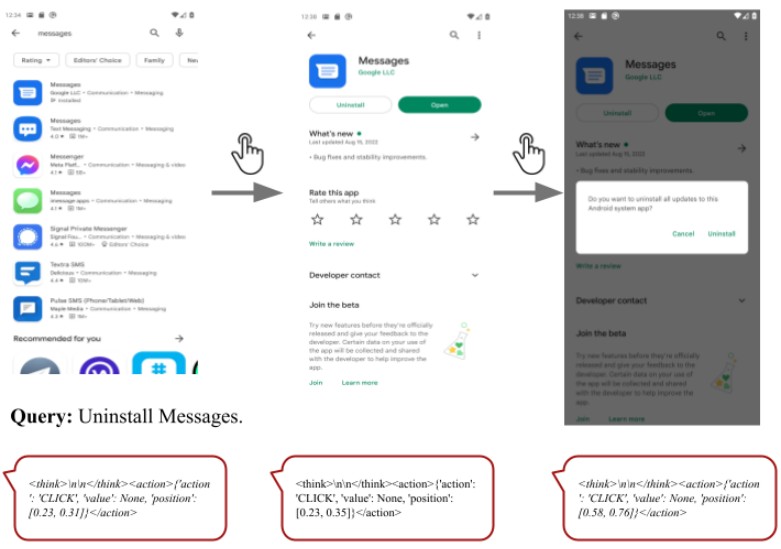

Query: Uninstall Messages.

*<think>\n\n</think><action>{'action': 'CLICK', 'value': None, 'position': [0.23, 0.31]}</action>*

<think>\n\n</think><action>{'action': 'CLICK', 'value': None, 'position': [0.23, 0.35]}</action>

*<think>\n\n</think><action>{'action': 'CLICK', 'value': None, 'position': [0.58, 0.76]}</action>*

Figure 25: TON applied to the AITZ out-of-distribution domain: the task is to uninstall the messaging apps. TON performs well without extensive reasoning.

Table 11: Counting example from SuperCLEVR [16]. Although the question is out-of-distribution, the performance of TON and vanilla GRPO differs significantly in their outputs. The output from vanilla GRPO is excessively lengthy (over 500 tokens), focusing more on reasoning rather than providing a direct answer. In contrast, TON delivers a concise response, effectively bypassing the lengthy reasoning process.

| **Counting examples from SuperCLEVR [16]** | | |
|---|---|---|
| | 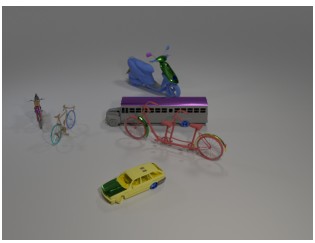 | 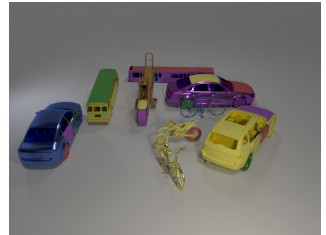 |
| Question | How many objects are either small green wagons or buses? | How many other objects are there of the same color as the matte car? |
| GRPO's outputs | "<think>, let's tackle this question step by step. The user is asking how many objects are either šmall green wagon-sör buses.̈ I need to check each object in the image provided., let's parse the image. There are six objects listed:1. A large blue rubber scooter with specific coordinates and rotation.2. A small yellow rubber car (wagon) with details.3. A small brown rubber bicycle.4. A large red rubber tandem bicycle.5. A small gray metal dirt bike.6. A large gray metal school bus.question is about two categories: small green wagons and buses. Let's break down each part.with šmall green wagons:̈ Looking at the list, there's a šmall yellow rubber carẅhich is a wagon. The color here is yellow, not green. So that doesn't fit. Are there any other small green wagons? The other objects are all different colors and types. The only green object mentioned is the first one, but it's a scooter, not a wagon. So no small green wagons exist in the scene., buses:̈ The sixth object is a l̈arge gray metal school bus.̈ Since the question says ", | <think>, let's see. The question is asking how many other objects are there of the same color as the m̈atte car.̇ First, I need to figure out which object is the matte car.at the image, each object has a color mentioned. Let me check each entry:1. A large yellow rubber SUV...2. A small blue metal bicycle...3. A large purple metal sedan...4. A large brown metal cylinder...5. A large yellow metal school bus...6. A small yellow metal bicycle...7. A large purple metal sedan...8. A small yellow rubber dirt bike...9. A large yellow metal school bus...10. A large purple metal articulated train..., the matte car isn't listed here. All the colors mentioned are either yellow, blue, purple, brown, or gray. The matte material isn't specified for any of them. Hmm, maybe m̈atter̈efers to the material being non-shiny, but in the given data, all materials are either metal or rubber. So perhaps the matte car is one of the rubber ones. But the problem says m̈atte,̈ so maybe it's a mistake, or maybe the user intended r̈ubberïnstead of m̈atte", |
| TON's outputs | <think>\n\n</think>
<answer>1</answer> | <think>\n\n</think>
<answer>3</answer> |

Q: As shown in the figure, AB parallel CD, straight line EF intersects AB at point E, intersects CD at point F, EG bisects angle BEF, and it intersects CD at point G, angle 1 = 50.0, then angle 2 is equal to ()

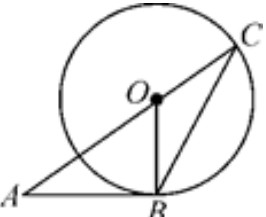

TON:

<think>\n\n</think><answer>65</answer>

GRPO:

<think>\nTo solve for angle 2, we need to use the properties of parallel lines and angle bisectors.\n\n1. Since AB is parallel to CD, and EF is a transversal, angle BEF and angle BFG are supplementary angles because they form a linear pair.\n2. Given that angle 1 is 50 degrees, and since EG bisects angle BEF, angle BEG must be half of angle BEF.\n3. Therefore, angle BEG = 90 - 50 = 40 degrees.\n4. Since EG bisects angle BEF, angle BEG = angle FEG = 40 degrees.\n5. Now, angle 2 is the same as angle FEG because they are corresponding angles formed by the transversal EF intersecting the parallel lines AB and CD.\n6. Therefore, angle 2 = 40 degrees.\n\n</think>\n\n<answer>\n40\n</answer>

Figure 26: Comparison between TON and vanilla GRPO in GeoQA

Q: As shown in the figure, AB ia tangent to circle O at point B, and the extended line of AO intersects circle O at point C. Connect BC, if angle A = 36.0, then angle C is equal to ()

TON:

<think>\n\n</think><answer>27</answer>

GRPO:

<think>\nTo solve for the measure of angle C, we need to use properties of circles and tangents.\n\n1. **Identify the given information:**\n  - AB is tangent to circle O at point B.\n  - Angle A is 36 degrees.\n  - We need to find the measure of angle C.\n2. **Use the property of the tangent:**\n  - Since AB is tangent to the circle at point B, the radius OB is perpendicular to AB. Therefore, angle OBA is 90 degrees.\n3. **Determine angle OAB:**\n  - In triangle OAB, the sum of the angles is 180 degrees.\n  - So, angle OAB + angle A + angle OBA = 180 degrees.\n  - Substituting the known values: angle OAB + 36 degrees + 90 degrees = 180 degrees.\n  - Simplifying, we get angle OAB = 54 degrees.\n4. **Identify angle COB:**\n  - Since OC is a radius and OB is also a radius, triangle OBC is isosceles with OB = OC.\n  - Therefore, angle OCB = angle OBC.\n5. **Determine angle COB:**\n  - The central angle COB subtends the same arc as the inscribed angle CBA.\n  - The measure of the central angle is twice the measure of the inscribed angle that subtends the same arc.\n  - So, angle COB = 2 * angle CBA.\n6. **Relate angles in triangle OBC:**\n  - In triangle OBC, the sum of the angles is 180 degrees.\n  - So, angle OCB + angle OBC + angle COB = 180 degrees.\n  - Since angle OCB = angle OBC, we can write: 2 * angle OCB + angle COB = 180 degrees.\n  - Substituting angle COB = 2 * angle CBA, we get: 2 * angle OCB + 2 * angle CBA = 180 degrees.\n  - Simplifying, we get: 2 * angle OCB + 2 * 36 degrees = 180 degrees.\n  - This simplifies to: 2 * angle OCB + 72 degrees = 180 degrees.\n  - Solving for angle OCB, we get: 2 * angle OCB = 108 degrees.\n  - Therefore, angle OCB = 54 degrees.\n\nSo, the measure of angle C is 54 degrees.\n</think>\n\n<answer>\n54\n</answer>

Figure 27: Comparison between TON and vanilla GRPO in GeoQA

Q: As shown in the figure, the diameter CD of circle O crosses the midpoint G of chord EF, angle DCF = 20.0, then angle EOD is equal to ()

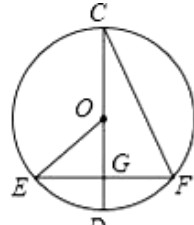

TON:

<think>\n\n</think><answer>40</answer>

GRPO:

<think>\nTo solve for the measure of angle EOD, we need to use properties of circles and angles.\n\n1. **Identify the given information:**\n  - \\( CD \\) is the diameter of the circle.\n  - \\( G \\) is the midpoint of the chord \\\\( EF \\\\).\n  - \\( \\angle DCF = 20^\\circ \\).\n\n2. **Understand the implications of the midpoint:**\n  - Since \\\\( G \\\\) is the midpoint of \\\\( EF \\\\), \\\\( CG \\\\) is perpendicular to \\\\( EF \\\\). This means that \\\\( \\angle CGE = 90^\\circ \\\\).\n\n3. **Use the inscribed angle theorem:**\n  - The angle subtended by an arc at the center of the circle is twice the angle subtended by the same arc at any point on the circumference.\n  - Therefore, \\\\( \\angle EOD \\\\) (the central angle) is twice \\\\( \\angle ECF \\\\) (the inscribed angle).\n\n4. **Determine the measure of \\\\( \\angle ECF \\\\):**\n  - Since \\\\( \\angle DCF = 20^\\circ \\\\) and \\\\( \\angle CGE = 90^\\circ \\\\), we can find \\\\( \\angle ECF \\\\) as follows:\n    \\\\[\n    \\angle ECF = 90^\\circ - 20^\\circ = 70^\\circ\n    \\\\]\n\n5. **Calculate \\\\( \\angle EOD \\\\):**\n  - Using the relationship between the central angle and the inscribed angle:\n    \\\\[\n    \\angle EOD = 2 \\times \\angle ECF = 2 \\times 70^\\circ = 140^\\circ\n    \\\\]\n\nThus, the measure of angle \\\\( EOD \\\\) is \\\\( 140^\\circ \\\\).\n\n</think>\n<answer>\n140\n</answer>

Figure 28: Comparison between TON and vanilla GRPO in GeoQA

Q: In the provided figure, the city government is making plans to construct an overpass with a height AB measuring 6.0. If angle ACB represents the angle between the slope AC of the overpass and the ground BC, and the sine of angle ACB is equal to 3.0/5.0, what is the length of slope AC?

TON:

<think>The length of the slope AC can be calculated using the Pytha... 3:4, we can set up a right triangle where the height (AB) is 6.0 unit... horizontal distance. Since the ratio is 3:4, we can use this to find the length of BC.

1. **Identify the ratio and relate it to the sides of the triangle:**

   - Let the length of AB be \( h \) and the length of BC be \( d \).

   - Given \( AB = 6.0 \) units, and the ratio is 3:4, we can express \( d \) in terms of \( h \):

   \[
   \frac{h}{d} = \frac{3}{4}
   \]

   Solving for \( d \):

   \[
   d = \frac{4}{3} \times 6.0 = 8.0 \text{ units}
   \]

2. **Apply the Pythagorean theorem to find the hypotenuse \( AC \):**

   \[
   AC = \sqrt{AB^2 + BC^2}
   \]

   Substituting the known values:

   \[
   AC = \sqrt{6.0^2 + 8.0^2}
   \]

   Calculate the squares:

   \[
   6.0^2 = 36.0 \quad \text{and} \quad 8.0^2 = 64.0
   \]

   Add these values:

   \[
   AC = \sqrt{36.0 + 64.0} = \sqrt{100.0} = 10.0 \text{ units}
   \]

Therefore, the length of the slope AC is 10.0 units.</think><answer>10</answer>

Figure 29: Comparison between TON and vanilla GRPO in GeoQA

Q: In the provided figure, the city government is making plans to construct an overpass with a height AB measuring 6.0. If angle ACB represents the angle between the slope AC of the overpass and the ground BC, and the sine of angle ACB is equal to 3.0/5.0, what is the length of slope AC?

**GRPO**

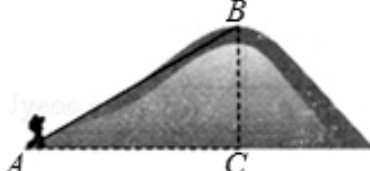

<think>

To solve for the length of the slope AC, we can use the given information about the slope and the vertical height. The slope AC is in the ratio 3.0:4.0, and the vertical height AB is 6.0 units.

1. **Identify the components of the right triangle:**

   - The vertical height (AB) is 6.0 units.

   - The ratio of the horizontal length (BC) to the vertical length (AB) is 3.0:4.0, so we can denote BC as $\frac{3}{4} \times AB$.

2. **Calculate BC:**

   - $BC = \frac{3}{4} \times 6.0 = 4.5$ units.

3. **Use the Pythagorean theorem to find AC:**

   - The Pythagorean theorem states $AC^2 = AB^2 + BC^2$.

   - Substitute the known values: $AC^2 = 6.0^2 + 4.5^2$.

   - $AC^2 = 36.0 + 20.25 = 56.25$.

   - $AC = \sqrt{56.25} = 7.5$ units.

Therefore, the length of the slope AC is 7.5 units.

</think>

<answer>

7.5

</answer>

Figure 30: Comparison between TON and vanilla GRPO in GeoQA

