# OpenReview forum: "Think or Not? Selective Reasoning via Reinforcement Learning for Vision-Language Models"
_NeurIPS.cc/2025/Conference — NeurIPS 2025 poster_

### Official Review · Reviewer_PYs6 · 2025-07-02

**Clarity:** 2
**Significance:** 2
**Originality:** 3
**Rating:** 4
**Confidence:** 3

**Summary:**

This paper introduces an adaptive training framework that lets Vision-Language Models flexibly produce either short or long chains of thought (CoT). The method uses a two-stage pipeline: **(1) supervised fine-tuning (SFT)** to establish two basic reasoning patterns followed by **(2) guided reinforcement preference optimization (GRPO)** to adjust CoT length to task demands. Experiments show that the framework simultaneously shortens unnecessary reasoning steps and improves overall performance.

**Questions:**

**Polished reviewer questions (with citations added)**

1. **Scope of application.**
   The framework is tested only on VLM reasoning tasks, yet recent length-adaptive methods in *text-only* settings—e.g., **ThinkLess** and **LASER**—report that LLM CoT traces can easily reach thousands of tokens, making compression arguably more urgent there. Could the authors clarify why the vision-language modality was chosen as the primary testbed, and whether the approach generalises to purely textual reasoning? ([arxiv.org][1], [arxiv.org][2])

2. **Source of performance gains.**
   The stated goal is to remove redundant reasoning steps; in principle this should *preserve* accuracy, not improve it. What specific mechanisms (e.g., better credit assignment in GRPO, reduced exposure bias) lead to the observed accuracy gains? Please provide ablations that disentangle efficiency improvements from accuracy improvements.

3. **Positioning against related work.**
   Please situate the contribution with respect to existing length-adaptive approaches such as **ThinkLess**, **LASER**, **AdaCoT**, and other “think-less” / difficulty-aware reward-shaping methods. What is the key novelty—reward design, the two-stage SFT + GRPO pipeline, or the adaptation to VLMs? A concise related-work discussion with explicit citations would help readers see the incremental contribution. ([1，2])

[1]: "ThinkLess: A Training-Free Inference-Efficient Method for Reducing Reasoning Redundancy"
[2]: "Learn to Reason Efficiently with Adaptive Length-based Reward Shaping"

**Ethical Concerns:**

["NO or VERY MINOR ethics concerns only"]

**Final Justification:**

The rebuttal addressed most of my concerns effectively. Consequently, I have revised my rating upwards to a 4.

**Limitations:**

yes

**Quality:**

2

**Strengths And Weaknesses:**

**Strengths**

* Delivers strong performance on all reported benchmarks.

**Weaknesses**

* The motivation for “over-dense” chains of thought is not fully substantiated. The evaluated tasks never exceed ≈ 1 k tokens of reasoning, so it is unclear why such CoTs are considered excessively long.

---

> ### Author Rebuttal · Authors · 2025-07-30
>
> > We appreciate that you found our method to have strong performance. We value your insights and hope our response addresses your comments.
> >
>
> Please find our detailed responses to your specific questions below. For clarity, we use the following notations:
>
> - W – Weakness
> - Q - Question
>
>
> # **W1: The motivation for “over-dense” chains of thought is not fully substantiated.**
>
> ---
>
> We would like to clarify that our motivation is **not** primarily aimed at optimizing over-dense chains of thought (i.e., making reasoning more efficient), but rather at addressing a more fundamental question:
> **"When should the model think?"** — i.e., identifying whether explicit reasoning is necessary.
>
> As shown in **Figure 2**, we conduct our motivation study on the AITZ benchmark, which does not exhibit excessively long CoT traces. Despite this, we observe that a non-trivial portion of cases (7.6%) still fail. More notably, **52.1% of the answers remain correct even without any reasoning**, and **14.5% are correct only when reasoning is skipped**. This implies that explicit reasoning is **not always required** for optimal performance.
> Further evidence and expanded analysis supporting this motivation are provided in **Appendix A**.
>
> We appreciate your insightful comment and agree that applying our framework to settings with long and potentially over-dense CoT is a valuable future direction. We plan to explore this in subsequent work.
>
> # **Q1: Why choose multimodal as the testbed? Can TON transfer to LLMs?**
>
> ---
>
> - We believe that vision tasks are more intuitive and can often be resolved at a glance, so it would be a good testbed to study think-or-not, while language-centric benchmarks are more focused on knowledge, which makes them difficult to articulate in detail. In contrast, text-based tasks typically provide more comprehensive information, thus would be better for long-cot. Visual tasks tend to be more redundant; often, a quick glance is sufficient to determine whether further thought is necessary, akin to human cognition.
> - TON is being generic, and we adapt our TON to the text-only datasets based on LLMs. Here, we give the experiments on the text-only tasks, and the results are as below. The results show that our TON reduces the length greatly while retaining the high accuracy. We have included these experiment settings in our revision.
>
> | gsm8k | pass@1 (↑) | think len (↓) |
> | --- | --- | --- |
> | Qwen2.5-1.5B-distill-deepseek-r1 | 78.7 | 939 |
> | grpo | 81.4 | 978 |
> | ton | **82.3** | **342** |
>
>
> # **Q2: What specific mechanisms of TON lead to accuracy gains? Considering that redundant reasoning, in principle, should preserve accuracy**
>
> ---
>
>
> **First**, we challenge the assumption that *redundant reasoning steps should preserve accuracy*. This is **not always true**. In fact, *saying more does not always mean saying it correctly*. As demonstrated in **Table 6** and **Figures 15 and 16**, lengthy but flawed reasoning processes can lead to incorrect answers by increasing the likelihood of hallucinations or logical errors.
>
> **Second**, we attribute our TON' effictiveness to two reasons.
>
> - **(i) TON effectively skips incorrect or redundant thinking processes, alleivating hallucinations.** **Fig. 2** shows that 52.1% of answers remained correct without the "think" process, and 14.5% were only correct when "think" was omitted. This suggests that explicit reasoning is not always necessary. **Sup. A.3** gives examples on GeoQA where the reasoning processes are either incorrect or redundant, leading to wrong answers. Our TON can skip these misleading reasoning steps and output answers directly and automatically.
> - **(ii) Introducing the non-think enlarge diversity on the distribution space of responses when GRPO sampling. Fig. 3** illustrates that TON can introduce empty thinking into the GRPO algorithm, thereby increasing response diversity within each group. Unlike previous works like DAPO emphasizes advantage distribution $A_i$ by dynamic sampling in the sparse reward space, our TON shifts the focus to the latent distribution space of responses $\pi_\theta(o_i|v,q)$, thus enhancing the diversity of the terms $\alpha, \beta$ in equation 4.
>
> Together, these mechanisms explain how TON improves accuracy—**not by blindly preserving all reasoning, but by selectively reasoning only when it adds value.**
>
>
> # **Q3: What is the novelty of TON compared to length-adapted related works?**
>
> ---
>
> **(i) TON v.s. length-adapted methods:** Length-adapted methods assume **reasoning is always needed** but optimize for shorter paths. Our TON tackles a more fundamental problem: **whether to think or not.** Figure 4(d) shows that the reasoning length of TON is similar to the vanilla GRPO.
>
> **(ii)** **Orthogonality and Compatibility.** TON’s *think-or-not* decision is **orthogonal** to length-adaptive methods. For instance, TON first determines whether reasoning is necessary; if it is, adaptive reasoning methods can then optimize the extent of that reasoning. This synergy holds great promise for future research.
>
>
> (iii) Lastly, we want to friendly remind that the works you suggested: Thinkless (21 May 2025), LASER (21 May 2025), AdaCoT (17 May 2025) mentioned above **were released after the NeurIPS2025 full paper submission (15 May 2025) deadline.** meaning that we are concurrent works. We would love to add a discussion of length-adaptive methods in related works in the revision.
>
> ---
> We appreciate your questions and comments very much. Please let us know for any further questions.

---

> > ### Author Response · Authors · 2025-08-04
> > **Kind Tips**
> >
> > Dear Reviewer PYs6,
> >
> > Thank you once again for your feedback and hope you have a good rest!
> >
> > Since the rebuttal period is halfway through, we would greatly appreciate it if you could review our response to ensure it adequately addresses your concerns. And we sincerely hope that you can reconsider our work in light of these clarifications. If you have any further comments, please do not hesitate to contact us.
> >
> > We greatly appreciate your selfless contributions to the community. Thank you for your time and consideration.
> >
> > Best regards,
> >
> > Authors of **Think or Not? Selective Reasoning via Reinforcement Learning for Vision-Language Models**

---

> > ### Comment · Reviewer_PYs6 · 2025-08-04
> >
> > Thank you for the clarifications and the additional experiments. The two explanations you provided for the accuracy improvement on TON are convincing. I am therefore raising my score to 4 and recommend accepting this paper.

---

> > > ### Author Response · Authors · 2025-08-04
> > > **Thanks for Your Time and Effort**
> > >
> > > Dear Reviewer PYs6,
> > >
> > > We are glad to receive your positive feedback. We greatly appreciate your selfless contributions to the community. We will include these discussions in the revision.
> > >
> > > Thank you again for your valuable suggestions and time!

---

### Official Review · Reviewer_bSRJ · 2025-07-03

**Clarity:** 4
**Significance:** 4
**Originality:** 4
**Rating:** 5
**Confidence:** 4

**Summary:**

This paper studies whether the model should conduct chain-of-thought reasoning or not in the multimodal LLM. The idea is quite simple and straightforward: replacing the thinking process with an empty thinking token. The motivation is the author found that in some cases, the thinking process might hurt the performance. Especially there are a high percentage where the model answered the questions incorrectly with the thinking mode on, but it can answer correctly without thinking mode. This motivated the author to study the thinking dropout approach. This is activated in the SFT stage. The GRPO algorithm is also updated with the thinking dropout process. This simple approach reduces the completion length by up to 90% and achieves quite good performance on counting (CLEVR) and Math(GeoQA) tasks.

**Questions:**

This is a good paper. I have several questions:

1. For the GRPO part, I didn't fully grasp how to update the GRPO algorithm to adapt the change of thinking dropout. Specifically, since the format reward is being added, how to handle the format reward for the sequence without thinking?
2. This approach should also be applicable to the text only LLM. I wonder what is the reason for studying this in the MLLM domain? Why not also studying this in the text only LLM domain and observe the performance improvement?
3. For the reverse thinking, how to verify the correctness of the thinking process?
4. An naive question for L141, why can't we just use the third party (closed source APIs) to generate the thinking process as the starting point?

**Ethical Concerns:**

["NO or VERY MINOR ethics concerns only"]

**Limitations:**

Yes

**Quality:**

4

**Strengths And Weaknesses:**

Strength:
1. The paper is well written. It is very easy to follow the whole paper. It is actually a joy to read this paper.
2. The proposed approach is very well motivated. Especially the Figure 2, which demonstrates the necessity for enabling the thinking dropout approach.
3. The approach is quite simple and straightforward. Frankly speaking, it just several line changes in the data processors. I am quite surprising about the simplicity of this approach and eager to try it out by myself.
4. The proposed approach achieves quite good performance in both efficiency and accuracy.

Weakness:
Frankly speaking, i didn't observe significant weakness of this work. The only thing that can improve is the clarity for how to update the GRPO. But the rest looks quite good to me. I would love to see this paper being accepted to this conference. It would be great if the author could also release the implementation.

---

> ### Author Rebuttal · Authors · 2025-07-30
>
> > We are very encouraged by your positive feedback and your recognizing the value of our proposed method! Below we address your concerns:
> >
> Please find our detailed responses to your specific questions below. For clarity, we use the following notations:
>
> - W – Weakness
> - Q - Question
>
>
>
>
> # W1: Would be great if the author could also release the implementation.
>
> ---
>
> Certainly! We value the impact of open-sourcing and are committed to releasing our models, code, and data to benefit the community.
>
> # Q1: How to handle the format reward for the sequence without thinking?
>
> ---
>
> Great question! Format specification is an important factor that influences model behavior. In our approach, we standardize both “thinking” and “non-thinking” modes into a unified format:
>
> `<think>reasoning process</think><answer>predicted answer</answer>`
>
> - For the **non-thinking format**, we use the structure `<think>\n\n</think>`, where the content inside `<think>` is replaced with `\n\n`.
> - In contrast, the **thinking format** retains the structure `<think>reasoning process</think>`, where the content explicitly represents the model’s reasoning process.
>
> This unified formulation enables supervised fine-tuning (SFT) to effectively incorporate thought dropout with minimal changes. Furthermore, it facilitates the transition to the GRPO stage by allowing reuse of the same format.
>
>
> # Q2: Why study this in the MLLM domain? Can we extend this in the LLM domain?
>
> ---
>
> Thank you for your suggestion regarding extending our study to the LLM domain.
>
> **Why we choice MLLM as a primary testbed:**
> - We believe that vision tasks are more intuitive and can often be resolved at a glance. In contrast, text-based tasks typically provide more comprehensive or professional information. Visual tasks tend to be more redundant; often, **a quick glance is sufficient to determine whether further thought is necessary**, akin to human cognition. There is why we choice MLLM to study this problem.
>
> **Extension of TON to the LLM Domain:**
> We appreciate your suggestion to extend our study to the LLM domain. In response, we have adapted our TON framework to the LLM benchmark GSM8K. Below, we present the corresponding experiments and results.
>
> The findings indicate that TON significantly reduces response length while maintaining high accuracy. This demonstrates the **generalizability of TON across different modalities.**
>
> We will include the experimental setup and incorporate further discussion on this extension in our revised submission.
>
>
> | GSM8K | Pass@1 (↑) | Avg. Think len (↓) |
> | --- | --- | --- |
> | Qwen2.5-1.5B-distill-deepseek-r1 | 78.7 | 939 |
> | grpo | 81.4 | 978 |
> | ton | **82.3** | **342** |
>
> # **Q3: For the reverse thinking, how to verify the correctness of the thinking process?**
>
> ---
>
> Good question! An intuitive approach to verifying the correctness of a thinking process is to *prepend it as a prefix and observe whether it enables the base model to generate the correct answer*. If the model succeeds under this condition, we can treat the thinking process as valid.
>
> However, based on our investigation, **the primary objective of the SFT stage is to instill format-following behavior**—such as the ability to either generate a complete reasoning trace or omit it entirely. Our intention is **not to intervene in the model’s behavior through external verification or enforcement**. This is why we adopt a reverse thinking strategy derived from the base model itself, without introducing a more powerful auxiliary model.
>
> We aim for the base model to self-explore during the GRPO stage, allowing us to observe the genuine gains achieved without relying on external guidance or supervision.
>
>
> # **Q4: Why not use third-party (closed-source APIs) to generate the thinking process as the starting point?**
>
> ---
>
> Yes, this is certainly possible. One could use third-party closed-source APIs to generate the thinking process simply by replacing the SFT training corpus with externally generated reasoning traces.
>
> However, **our focus is on enabling the model to develop its own reasoning abilities without external assistance**. By doing so, we ensure that *any improvements observed during the GRPO stage stem from the model itself, rather than being distilled from a stronger external model.* This setup allows for a clearer evaluation of the model's intrinsic capacity for reasoning and self-improvement.
>
> ---
>
> We appreciate your questions and comments very much. Please let us know for any further questions.

---

> > ### Author Response · Authors · 2025-08-04
> > **Kind Tips**
> >
> > Dear Reviewer bSRJ,
> >
> > Thank you once again for your valuable and positive feedback! Hope you have a good rest!
> >
> > Since the rebuttal period is halfway through, we would greatly appreciate it if you could review our response to ensure it adequately addresses your concerns. If you have any further comments, please do not hesitate to contact us.
> >
> > We greatly appreciate your selfless contributions to the community. Thank you for your time and consideration.
> >
> > Best regards,
> >
> > Authors of **Think or Not? Selective Reasoning via Reinforcement Learning for Vision-Language Models**

---

> ### Comment · Area_Chair_p76H · 2025-08-08
>
> Dear Reviewer bSRJ,
>
> This is the last call for author-reviewer discussion, which will ends on Aug 8. Could you please read authors' rebuttal and confirm whether your concerns have been addressed asap? Thank you.
>
> Best,
>
> AC

---

### Official Review · Reviewer_g1TJ · 2025-07-03

**Clarity:** 3
**Significance:** 2
**Originality:** 3
**Rating:** 3
**Confidence:** 4

**Summary:**

This paper introduces TON, a two-stage training framework for vision-language models (VLMs) that instills selective reasoning ability by combining supervised fine-tuning with “thought dropout” and reinforcement learning. TON, evaluated on multiple benchmarks, substantially reduces reasoning trace length, while sometimes improving accuracy over GRPO. The approach demonstrates strong out-of-distribution generalization.

**Questions:**

Could the authors clarify the statement in L217 regarding why directly applying GRPO leads to a zero coordinate reward? The explanation is unclear.

**Ethical Concerns:**

["NO or VERY MINOR ethics concerns only"]

**Final Justification:**

I maintain my BR rating. Using purely random dropout to decide whether to remove reasoning traces is unreasonable, since it may harm the model’s reasoning capability when the reasoning trace for a complex question is dropped. Besides, the provided evidence, showing that random dropout outperforms difficulty-aware dropout, comes from a single dataset and is neither comprehensive nor general enough to rule out bias. Without substantial and diverse experiments to prove this, the methodology remains flawed, and I cannot recommend it for acceptance.

**Limitations:**

yes

**Quality:**

2

**Strengths And Weaknesses:**

[Strength]
1. The idea to equip VLMs with selective reasoning is practical and well-motivated.
2. TON achieves a significant reduction in average reasoning length without sacrificing performance, surpassing vanilla GRPO.
3. The reward design with continuous matching is straightforward yet effective across benchmarks.


[Weaknesses]
1. Applying random dropout of reasoning traces during SFT may harm the model’s reasoning on complex queries, such as GeoQA, where reasoning steps are critical.
2. This paper lacks a detailed analysis of which query types the model learns to skip reasoning for after training. Providing quantitative insights into these categories would clarify the model’s selectivity and failure modes.

---

> ### Author Rebuttal · Authors · 2025-07-30
>
> > Thank you for recognizing the value of our work and for providing constructive and thorough feedback. We value your insights and hope our response addresses your comments.
> >
>
> Please find our detailed responses to your specific questions below. For clarity, we use the following notations:
>
> - W – Weakness
> - Q - Question
>
>
> # **W1: Rationale for Random Dropout of Reasoning Traces**
>
> ---
>
>
> Thank you for your comments on the complex reasoning case, we also value this most. We appreciate the opportunity to discuss it in greater depth.
>
> **1. Does adding the reasoning difficulty supervision make it better?**
>
>
>
> - **By Answer Correctness**:
>  We tried a direct approach to classify each sample's difficulty based on *whether the base VLM answers it correctly*. We then drop thoughts for only the “easy” samples (those answered correctly by the base VLM). To investigate this, we implemented a difficulty-aware dropout strategy and compared its accuracy under our TON training framework.
>
> | Qwen2.5-VL-3b (GeoQA) | TON acc(↑) |
> | --- | --- |
> | Difficulty-aware dropout | 50.5 |
> | Random dropout (ours) | **51.0** |
>
> Somewhat surprisingly, the accuracy of the difficulty-aware dropout version is similar to, and even slightly lower than, our proposed random dropout method. This outcome may be attributed to the challenge of precisely defining “task difficulty”. Human-defined heuristics introduce potential noise and risk interfering with model behavior.
>
> Therefore, we adopt **a conservative and principled approach**: we treat all samples equally, allowing the model to acquire format-following skills during SFT and to self-explore and learn effectively during the GRPO stage. This strategy ensures that the model’s improvements are intrinsic and not confounded by external definitions or interventions.
>
> **2. Despite Random Dropout in SFT, the Model Adapts to Task Difficulty in the RL Stage**
>
> As shown in Supplementary Section A.1, even with random thought dropout during SFT, our TON framework enables the model to adapt its skip-thinking behavior during GRPO based on task difficulty and reasoning complexity.
> We analyze performance across a range of benchmarks—including 3D object counting, GUI-based mobile agent navigation, and geometric math reasoning—spanning a spectrum of task difficulties. Task difficulty is assessed both qualitatively (via intuitive reasoning complexity) and quantitatively (by base VLM zero-shot accuracy), grouped as:
>
> - 60-100 (easy)
> - 40-60 (medium)
> - 0-40 (hard).
>
> The results of the final skip-ratios and our TON accuracy are as follows:
>
> | Types | zero-shot acc | skip ratio | TON acc(↑) |
> | --- | --- | --- | --- |
> | [easy] Counting | 64.0 | 98% | 98.5 |
> | [middle] AITZ | 58.0 | 79% | 74.0 |
> | [hard] GeoQA | 38.0 | 70% | 51.0 |
>
> Our findings indicate that our **TON maintains a high skip ratio for easy counting queries, allowing it to bypass unnecessary thinking processes, while keeping a low skip ratio for harder, more professional, knowledge-intensive questions to fully utilize its thinking capabilities.**
>
> **3. Possible future extension:**
> We acknowledge that the skip-thinking mechanism may introduce the risk of the model bypassing necessary complex reasoning steps, and we plan to address it in future work.
> **(i)** A potential direction is to incorporate auxiliary scores such as visual perplexity (PPL) scores as a proxy for estimating task difficulty.
> **(ii)** Another possibility is to evaluate whether the model can answer a question correctly both with and without the reasoning trace. If so, the reasoning may be redundant and thus safely skippable.
>
>
> # **W2: Would be better to provide quantitative insights into the skip ratios and query types.**
>
> ---
>
> Thank you for your valuable suggestion. To address this, we categorize query types based on their originating benchmarks. For example, we identify query types such as:
>
> - **Counting** (e.g., "How many [objects]?")
> - **Mobile Agent** (e.g., "Download a music app")
> - **Math Reasoning** (e.g., "What is the degree of the angle?")
>
> These query types naturally differ in their reasoning complexity. To quantify this, we assess task difficulty using the base VLM’s accuracy in a zero-shot setting:
>
> - **Easy**: 60–100%
> - **Medium**: 40–60%
> - **Hard**: 0–40%
>
> We then analyze the model’s skip-thinking behavior during the GRPO stage and observe how the skip ratio correlates with query type and difficulty.
>
> | Types | zero-shot acc | skip ratio | TON acc(↑) |
> | --- | --- | --- | --- |
> | [easy] Counting | 64.0 | 98% | 98.5 |
> | [middle] AITZ | 58.0 | 79% | 74.0 |
> | [hard] GeoQA | 38.0 | 70% | 51.0 |
>
> Our findings indicate that our **TON maintains a high skip ratio for easy counting queries, allowing it to bypass unnecessary thinking processes, while keeping a low skip ratio for harder, more professional, knowledge-intensive questions to fully utilize its thinking capabilities.**
>
> # **Q1: Why does directly applying GRPO lead to a zero coordinate reward in L217**
>
> ---
>
> There is because in AITZ benchmark, we require the model to output well structural action format, must adhere to JSON (lines 222-224),`{action: 'ACTION_TYPE', value: 'ELEMENT', position: '[x,y]'}`, which is more complex than the math problems with a numerical answer (e.g., 5).
>
> We find that open-source models without SFT format following cannot directly follow such complex instructions, making it difficult to extract answers and determine consistency with the ground truth, resulting in large failures. While RL stage is not good for format following and is better for generalization, thus without SFT, direclty GRPO cannot obtain reward to support training.
>
> ---
>
> We appreciate your questions and comments very much. Please let us know for any further questions.

---

> > ### Author Response · Authors · 2025-08-04
> > **Kind Tips**
> >
> > Dear Reviewer g1TJ,
> >
> > Thank you once again for your feedback and hope you have a good rest!
> >
> > Since the rebuttal period is halfway through, we would greatly appreciate it if you could review our response to ensure it adequately addresses your concerns. And we sincerely hope that you can reconsider our work in light of these clarifications. If you have any further comments, please do not hesitate to contact us.
> >
> > We greatly appreciate your selfless contributions to the community. Thank you for your time and consideration.
> >
> > Best regards,
> >
> > Authors of **Think or Not? Selective Reasoning via Reinforcement Learning for Vision-Language Models**

---

> ### Comment · Reviewer_g1TJ · 2025-08-05
> **Official Comment by Reviewer g1TJ**
>
> Thanks for the detailed rebuttal. Most of the concerns are addressed except W1. The experiments show random dropout actually outperforms difficulty-aware dropout (51.0 vs 50.5), suggesting that defining difficulty via noisy labels (answer correctness) may not be optimal. Thus, the current version could be improved by a more proper definition of difficulty (instead of directly using noisy labels). I decide to maintain a borderline negative rating.

---

> > ### Author Response · Authors · 2025-08-08
> >
> > Dear Reviewer g1TJ,
> >
> > I hope this message finds you well. As the discussion phase nears its end, with just **one day remaining**, I would like to ensure that we have thoroughly addressed all your concerns, especially our heuristic-free dropout strategy. If there are any additional points of feedback you would like us to consider, please let us know. Your insights are invaluable to us, and we’re eager to address any remaining issues to improve our work.  We sincerely hope that you could reconsider our work in light of these clarifications.
> >
> > Thank you for your time and effort in reviewing our paper.
> >
> > Authors of Submission 15147

---

> ### Author Response · Authors · 2025-08-05
> **A Clarification note on our heuristic-free dropout strategy**
>
> Thank you so much for digging into our difficulty‐aware dropout ablation. We’d like to clarify two key points:
>
> 1. **Ablation, not contribution.**
> The “difficulty-aware” dropout—we drop thoughts only on examples the base VLM answers correctly—was included solely as an ablation study at your suggestion. It is **not** our core contribution. Its purpose was to demonstrate that *even a hand-crafted, noisy difficulty signal cannot match the performance of our simpler strategy*.
> 2. **Core innovation: random dropout.**
> Our main idea is that **randomly dropping reasoning traces during SFT** lets the model learn pure format-following, then autonomously decide when to think or skip during the RL stage, **all without any manual difficulty labels**. The 0.5 gain (51.0 vs. 50.5) shows that *human-defined “easy/hard” heuristics often introduce noise and can actually hinder performance compared to a fully random approach*.
>
> We hope this clears things up: our strength really comes from **doing away with handcrafted difficulty definitions**. If you have any thoughts, suggestions, or concerns about this “heuristic-free” angle of TON, we’d be grateful to hear them.

---

### Official Review · Reviewer_vGCC · 2025-07-05

**Clarity:** 2
**Significance:** 2
**Originality:** 2
**Rating:** 4
**Confidence:** 3

**Summary:**

Considering current vision-language model using RL to enhance reasoning ability causing redundant response, the author is motivated to enable models to learn selective reasoning—knowing when to reason and when to skip it—rather than indiscriminately generating reasoning traces, this paper proposes TON, a two-stage method: SFT+GRPO. In the first stage, the author teaches the model to skip reasoning appropriately by prompting model to self-generate reasoning trace given image, question, answer and later randomly replacing reasoning traces with empty “\n\n” using to construct dataset. In the second stage, the author designs discrete matching and continuous matching reward to optimize model’s selective reasoning ability. The primary contribution of this work lies in introducing “selective reasoning” as a trainable capability for vision-language models, significantly reducing redundancy in model responses.

**Questions:**

（1）Clarification of the Two-Stage Design Rationale (SFT + GRPO)
The paper proposes a two-stage framework for selective reasoning, beginning with SFT using thought dropout and followed by a GRPO-based reinforcement learning stage. However, the motivation for structuring the method in this specific two-stage manner is not fully explained. In particular, it remains unclear whether random dropout may lead to skip complex reasoning after SFT. Could the authors provide more conceptual reasoning or design analysis supporting this choice? Including additional ablation experiments comparing dropout strategies would help clarify this point. Addressing this question could significantly improve the paper’s Clarity score.

（2）Generalizability of Reward Design in the GRPO Stage
Does the reward design satisfy the case when outcomes may not be easily judged by binary or discrete metrics? It may be more convincible to comment on the adaptability of reward design to other situation? For instance, would the current selective reasoning policy be viable in tasks lacking deterministic rewards? If the approach can be extended or generalized to such settings, it would enhance the broader applicability of selective reasoning and justify raising the Significance rating.

（3）Deeper Analysis of “Without Think but Correct” Cases in Ablation
In the early-stage ablation where responses with and without “think” are compared, a significant proportion of cases are correctly answered without reasoning while being incorrect with it. This is intriguing but underexplored. Could the authors analyze these cases more deeply—are the errors in the “think” version due to hallucinated reasoning, misaligned chains of thought, or memory interference? Furthermore, after applying TON, how do such cases behave? Are they handled better by skipping reasoning, and if so, what insights can we draw about the model’s internal reasoning reliability? A targeted analysis would shed light on whether selective reasoning improves not just efficiency but also robustness. This could positively impact the Clarity score if addressed with empirical evidence or case studies.

（4）The specificity of vision-language tasks compared to only language task
The proposed method does not appear to differentiate significantly between pure-text and vision-language scenarios. While the perspective of exploring reasoning length in vision-language tasks is novel, the method itself does not explicitly leverage the unique characteristics of vision-language reasoning. Could the authors consider incorporating task-specific aspects of vision, or provide a comparison that highlights how this approach differs from existing methods for reducing reasoning length in pure language models? Addressing this could support an increase in both the Significance and Originality scores.

**Ethical Concerns:**

["NO or VERY MINOR ethics concerns only"]

**Limitations:**

Yes

**Quality:**

3

**Strengths And Weaknesses:**

This paper presents a technically sound framework with comprehensive experiments across multiple tasks and model scales. The use of thought dropout and reinforcement learning is well-motivated and evaluated through thorough ablations and clear metrics. However, the construction of dataset by randomly dropout reasoning trace may lead to model skip some complex reasoning after SFT which can’t be optimized effectively by GRPO with task-specific two kinds of reward designs.
The paper is well-organized and clearly communicates its motivation and methodology. By implementing ablation experiments of with/without thinking on AITZ task, the paper introduces its motivation of selective reasoning. And the experiment part, especially the ablation of 3 questions well suits the intuitive consideration of the “think or not” mechanism. The only slight shortcoming lies in the implementation of selective reasoning — the rationale for adopting a two-stage approach requires further clarification.
The work addresses reducing reasoning overhead in vision-language models by reinforcing selective reasoning. This has practical value for efficient inference in real-world applications.
The idea of teaching models when to reason, rather than how, is a novel and insightful shift. Although similar ideas exist in adaptive computation, the integration into multimodal RL training is new and well-executed.

---

> ### Author Rebuttal · Authors · 2025-07-30
>
> > We are very encouraged by your positive response! Below we have addressed your questions:
> >
>
> Please find our detailed responses to your specific questions below. For clarity, we use the following notations:
>
>
> - W – Weakness
> - Q - Question
>
>
>
>
> # **Q1: Rationale of Two-stage.**
>
> ---
>
> The two-stage framework originated from our exploratory experiments in **Section 5.4--Emprical Verfication of SFT Significance in TON**.
> - We initially attempted to utilize only the GRPO stage without the preceding SFT stage: **(1)** *hybrid prompts where prompt the models to skip think if it is confidence enough* and **(2)** *Non-think rewards give reward 1 if it skip the think*. Unfortunately, these attempts did not induce skipping;
> - Fig. 5d shows a skip ratio of 0, and the training curve in Appendix. G indicates that non-think rewards also remained at 0. We analyzed this phenomenon and attributed it to the *VLMs defaulting to a conservative, fully reasoning approach*, resulting in a nearly 100% think rate.
>
> This motivated us to **inject the non-think prior during the SFT stage using thought dropout**, which we found necessary to break this bias toward full reasoning.
>
> # **Q1: Rationale of Random dropout.**
>
> ---
>
> Thank you for raising this question. Below, we outline the motivation and supporting analysis behind our use of random dropout for reasoning traces:
>
> **1. Random vs. Difficulty-aware Dropout**
> We also experimented with a difficulty-aware dropout strategy, where reasoning is dropped only for "easy" samples (those correctly answered by the base VLM). The comparison on GeoQA is as follows:
>
> | Method                   | TON Accuracy (↑) |
> |--------------------------|------------------|
> | Difficulty-aware dropout | 50.5             |
> | Random dropout (ours)    | **51.0**         |
>
> Surprisingly, random dropout performs slightly better. This suggests that hand-crafted heuristics for task difficulty may *introduce noise or unintended bias, potentially interfering with the learning process*. **Random dropout offers a simpler, unbiased alternative that generalizes well across tasks.**
>
> **2. Adaptive Behavior in GRPO**
>
> Despite random dropout during SFT, our model learns to adaptively skip or retain reasoning in GRPO based on task difficulty. For example:
>
> | Task Type      | Zero-shot Acc | Skip Ratio | TON Acc (↑) |
> |----------------|----------------|-------------|--------------|
> | [Easy] Counting | 64.0           | 98%         | 98.5         |
> | [Medium] AITZ   | 58.0           | 79%         | 74.0         |
> | [Hard] GeoQA    | 38.0           | 70%         | 51.0         |
>
> TON maintains a high skip ratio for easy tasks while preserving reasoning for more complex queries, demonstrating that **the model internalizes task difficulty and adjusts its behavior accordingly—even without explicit difficulty labels in SFT.**
>
> In summary, random dropout is a simple yet effective mechanism to encourage flexible, adaptive reasoning while avoiding the heuristic-based supervision. We will expand this discussion in the revised paper.
>
>
>
>
> # **Q2: Can TON be scalable to tasks without binary or deterministic rewards?**
>
> ---
>
> TON primarily addresses **selective reasoning in VLMs** by introducing a generic thought-dropout method, due to its generic and concise method, it do not add any regularization on the reward definitions. Thus, the TON framework can still be adapted to uncertain environments in the following ways:
>
> 1. **Adapting reward beyond binary/discrete metrics:** In such cases, we can still define formal reward instead of binary metrics, such as in Mobile navigation, regrading the action grounding, we could use the absolute L1 distance between click coordinate and box as continuous reward measurement.
> 2. **For these tasks without deterministic rewards:** This being an open challenge in RL training, a potential solution could be training a critic model or adopting powerful models to predict rewards to supervise training.
>
> But we want to emphasize that as TON does not introduce explicit reward regularization; thus, TON could be flexibly adapted to these scenarios. We would like to explore this extension in future work.
>
> # Q3: Deeper Analysis of think and How TON improve it
>
> ---
>
> Thanks for raising insightful question regarding the our finding. Follow your suggestion, we analyze 10 examples in Appendix G.8 and Supp. A.3 with the outputs of zero-short, grpo and our TON on 3 tasks.  We have the following obseravations:
>
> - **The model may generate hallucinated reasoning that leads to incorrect answers.**
>    As shown in **Table 10**, even on simple 3D counting tasks (e.g., "How many items are in the picture?"), the model may produce meaningless or incorrect reasoning steps, ultimately resulting in wrong answers.
>
> - **TON exhibits greater robustness with respect to the presence or absence of reasoning.** As shown in **Figures 15 and 16**, TON is capable of providing correct answers even without an explicit reasoning process. We attribute this to two key factors:
>
> **(i)** TON can selectively skip reasoning steps, thereby **reducing the likelihood of hallucinations or logical errors.**
>
> **(ii)** For many VLMs, this suggests that **generating a correct answer along with a valid reasoning trace is more difficult than generating the answer alone.**
>
>
>
> **How TON improve this issue?**
> - **TON can skip the reasoning process to alleviate incorrect reasoning, allowing it to provide direct answers for easier questions.** In Table 5, models outputs a lengthy but incorrect reasoning process (~1k), which hinders the model's ability to answer correctly. In contrast, TON directly provides the correct answer with minimal reasoning.
> - Insights about the model’s internal reasoning reliability: **for simpler questions, skipping the reasoning process reduces the probability of errors, offering a more efficient pathway to correct answers.**
>
> These analyses provide a deeper understanding of the behavior and advantages of our TON framework. We will incorporate this discussion into the revision.
>
>
> # Q4: Differentiate TON from existing length adaptive reasoning methods in LLM
>
> ---
>
>
> We thank you for raising such insightful comments.
>
> Firstly, it is necessary to clarify the key differences between our TON to prior adaptive reasoning methods.
>
> (i) **How to think v.s. When to think**: Adaptive reasoning methods assume reasoning is **always needed but optimize for shorter paths**. Our TON tackles a more fundamental problem: **whether to think or not**. Figure 4(d) shows that the reasoning length of TON for those unskipped questions is still similar to the vanilla GRPO.
>
> (ii) **Orthogonality and Compatibility**. TON’s think-or-not decision is **orthogonal to** length-adaptive methods. For instance, TON first determines whether reasoning is necessary; if it is, adaptive reasoning methods can then optimize the extent of that reasoning. This synergy holds great promise for future research.
>
> In the future, we would like to include more visual-centric designs, such as extend our ‘dropout’ to visual inputs, let model to skip redundant visual tokens as well.
>
> ---
>
> We appreciate your questions and comments very much. Please let us know for any further questions.

---

> > ### Author Response · Authors · 2025-08-04
> > **Kind Tips**
> >
> > Dear Reviewer vGCC,
> >
> > Thank you once again for your feedback and hope you have a good rest!
> >
> > Since the rebuttal period is halfway through, we would greatly appreciate it if you could review our response to ensure it adequately addresses your concerns. If you have any further comments, please do not hesitate to contact us.
> >
> > We greatly appreciate your selfless contributions to the community. Thank you for your time and consideration.
> >
> > Best regards,
> >
> > Authors of **Think or Not? Selective Reasoning via Reinforcement Learning for Vision-Language Models**

---

> > ### Comment · Reviewer_vGCC · 2025-08-04
> >
> > Thank you for your reply. I read the review and most of my concerns have been addressed.

---

> > > ### Author Response · Authors · 2025-08-04
> > > **Thanks for your time and efforts!**
> > >
> > > Thank you very much for taking the time to review our rebuttal and for sharing your decision.
> > >
> > > We greatly appreciate your thoughtful feedback and helpful suggestions. We're glad to hear that our response addressed your concerns. If you feel that our rebuttal adequately address your earlier concerns, we would be truly grateful if you might consider adjusting your score accordingly.
> > >
> > > We sincerely appreciate your time and constructive feedback throughout this process.

---

> > > ### Author Response · Authors · 2025-08-08
> > >
> > > Dear Reviewer vGCC,
> > >
> > > Thank you for your positive feedback! As the discussion phase nears its end, with just **one day remaining**, I want to ensure that we have thoroughly addressed all your concerns.
> > >
> > > We noticed you mentioned that “most of” the issues have been resolved. If there are any remaining doubts or specific points that still need clarification, please let us know—understanding these outstanding matters will help us further improve our work.
> > >
> > > We remain committed to addressing all feedback to ensure the highest quality of our submission. If you have no other concerns, would you consider increasing your rating of our service? We would be very grateful if you did. We’re also happy to provide any additional details or explanations you may need. Thank you again for your time and invaluable insights throughout the review process.
> > >
> > > Authors of Submission 15147

---

### Official Review · Reviewer_WjY5 · 2025-07-05

**Clarity:** 3
**Significance:** 2
**Originality:** 2
**Rating:** 3
**Confidence:** 4

**Summary:**

This paper proposes a method to enable selective reasoning for VLMs. The
proposed method works by using a simple "random thought dropout" approach during
supervised fine-tuning and incentivize "selective thought dropout" during RL
post-training (e.g., GRPO). Experiments on models from 3B to 7B demonstrate the
effectiveness of the proposed method.

**Questions:**

see weaknesses

**Ethical Concerns:**

["NO or VERY MINOR ethics concerns only"]

**Final Justification:**

Please see the comments for the rebuttal.

**Limitations:**

yes

**Quality:**

2

**Strengths And Weaknesses:**

# Strengths

* The studied problem is interesting. The motivation is clear.

* The overall writing of this paper is good.

* Both out-of-distribution data and in-distribution data is involved in the evaluation.

# Weaknesses

* The proposed thought dropout mechanism dring SFT is applied randomly without
considering task difficulty or reasoning necessity. This could interfere with
the model’s reasoning capability and the ability to calibrate when reasoning is needed. It is suggested to
add more discussion about this potential drawback (although the RL stage might be able to fix, it could still introduce potential negative effects). More principled approaches,
such as difficulty-aware or confidence-based dropout might be also helpful.
These adaptive approaches are not explored or discussed. In addition, the optimal dropout rate is likely to depend on the difficulty distribution of the SFT dataset. A discussion on how to align dropout rates with dataset characteristics is missing and would improve the soundness of the proposed method.

* Although the scope of this paper is limited to VLMs, the method itself is
generic and could easily be applied to text-only LLMs. However, there are many
existing works on adaptive reasoning on LLMs, such as those mentioned in the
related work section, but none are included in the experimental comparison.
This limits the clarity of the paper’s contribution comparing to prior works.

* All results are based solely on Qwen2.5-VL 3B and 7B. There is no evaluation
on other model families (e.g., InternVL3 and LLaMA 3.2 vision) or larger models.
It is unclear whether the method scales to stronger models or transfers across
different VLM architecturesa, which weakens the generality of the proposed
method and the contribution. Is the injection of the thought dropout harder for
larger models as they have stronger reasoning capability and better inherent
calibration of when reasoning is needed?

---

> ### Author Rebuttal · Authors · 2025-07-30
>
> > We appreciate your valuable feedback and are glad to hear that you found the selective reasoning problem interesting. We value this rebuttal opportunity to clarify issues and address your concerns.
> >
> Please find our detailed responses to your specific questions below. For clarity, we use the following notations:
>
> - W – Weakness
> - Q - Question
>
>
>
> # **W1: Consider Thought Dropout with task difficulty.**
>
> ---
>
> Thank you for emphasizing the importance of task difficulty—this is indeed a crucial factor, we also value this most, we appreciate the opportunity to discuss it in greater depth.
>
> **1. How to measure Task Difficulty?**
>
>
> - **By Answer Correctness**:
>  A more direct approach is to classify each sample based on *whether the base VLM answers it correctly*. We then drop thoughts for only the “easy” samples (those answered correctly by the base VLM). To investigate this, we implemented a difficulty-aware dropout strategy and compared its accuracy under our TON training framework.
>
> | Qwen2.5-VL-3b (GeoQA) | TON acc(↑) |
> | --- | --- |
> | Difficulty-aware dropout | 50.5 |
> | Random dropout (ours) | **51.0** |
>
> Somewhat surprisingly, the accuracy of the difficulty-aware dropout version is similar to, and even slightly lower than, our proposed random dropout method. This outcome may be attributed to the challenge of precisely defining “task difficulty”. Human-defined heuristics introduce potential noise and risk interfering with model behavior.
>
> Therefore, we adopt **a conservative and principled approach**: we treat all samples equally, allowing the model to acquire format-following skills during SFT and to self-explore and learn effectively during the GRPO stage. This strategy ensures that the model’s improvements are intrinsic and not confounded by external definitions or interventions.
>
> **2. Despite Random Dropout in SFT, the Model Adapts to Task Difficulty in the RL Stage**
>
> As shown in Supplementary Section A.1, even with random thought dropout during SFT, our TON framework enables the model to adapt its skip-thinking behavior during GRPO based on task difficulty and reasoning complexity.
> We analyze performance across a range of benchmarks—including 3D object counting, GUI-based mobile agent navigation, and geometric math reasoning—spanning a spectrum of task difficulties. Task difficulty is assessed both qualitatively (via intuitive reasoning complexity) and quantitatively (by base VLM zero-shot accuracy), grouped as:
>
> - 60-100 (easy)
> - 40-60 (medium)
> - 0-40 (hard).
>
> The results of the final skip-ratios and our TON accuracy are as follows:
>
> | Types | zero-shot acc | skip ratio | TON acc(↑) |
> | --- | --- | --- | --- |
> | [easy] Counting | 64.0 | 98% | 98.5 |
> | [middle] AITZ | 58.0 | 79% | 74.0 |
> | [hard] GeoQA | 38.0 | 70% | 51.0 |
>
> Our findings indicate that our **TON maintains a high skip ratio for easy counting queries, allowing it to bypass unnecessary thinking processes, while keeping a low skip ratio for harder, more professional, knowledge-intensive questions to fully utilize its thinking capabilities.**
>
> **3. Possible future extension:**
> We acknowledge that the skip-thinking mechanism may introduce the risk of the model bypassing necessary complex reasoning steps, and we plan to address it in future work.
>
> **(i)** a potential direction is to incorporate auxiliary score such as visual perplexity (PPL) scores as a proxy for estimating task difficulty.
>
> **(ii)** Another possibility is to evaluate whether the model can answer a question correctly both with and without the reasoning trace. If so, the reasoning may be redundant and thus safely skippable.
>
> # **W1: Discussion of dropout rates with dataset characteristics.**
>
> ---
>
> We have conducted a dataset-aware analysis of dropout behavior as part of our experimental investigation. Specifically,
>
> - [**Same dataset, different ratio**] In **lines 255-263**, we present the ablation study and insights into selecting dropout rates of 20%, 50%, and 80%. Our findings indicate **a correlation between the dropout ratio and the zero-shot performance across the datasets.**
>
> - [**Different dataset, different ratio**] In **Appendix G.5 and G.6**, we analyze the training curves of these dropout ratios across datasets of varying reasoning difficulties.
> we observe that, **despite differing dropout ratios, TON consistently shows an increasing skip ratio as training progresses**. Notably, the lower dropout ratios demonstrate a rapid increase in skip rates, while the higher ratios remain relatively stable throughout training. TON can dynamically adjust according to reward signals—decreasing the dropout ratio when performance is high and increasing it when performance declines. Thus, we recommend starting with a lower dropout probability for further investigation, such as a 20% ratio for more difficult tasks (i.e., zero-shot accuracy < 40) and an 80% ratio for easier tasks (i.e., zero-shot accuracy > 60).
>
> We thanks for your suggestions. These insights enhance our deeply understanding of TON, and we will highlight them in our revision.
>
>
>
> # **W2: Would be better to have Discussion between Adaptive reasoning methods in LLMs and TON.**
>
> ---
>
> Thanks for your suggestion and for mentioning our work with these adaptive reasoning methods.
>
> Firstly, it is necessary to clarify the key differences between our TON to prior adaptive reasoning methods.
>
> **(i) How to think *v.s.* When to think:** Adaptive reasoning methods assume **reasoning is always needed** but optimize for *shorter* paths. Our TON tackles a more fundamental problem: **whether to think or not.** Figure 4(d) shows that the reasoning length of TON for those unskipped questions is still similar to the vanilla GRPO.
>
> **(ii)** **Orthogonality and Compatibility.** TON’s *think-or-not* decision is **orthogonal** to length-adaptive methods. For instance, TON first determines whether reasoning is necessary; if it is, adaptive reasoning methods can then optimize the extent of that reasoning. This synergy holds great promise for future research.
>
> **(iii)** We further **include the representative adaptive reasoning method DAPO** [1], which apply the adaptive reasoning with dynamic sampling and the normalziation of the token numbers within the group; We apply it on VLMs for comparison on GeoQA, and the results are as below: Despite both variants yield signifcant improvement over original GRPO, our TON with minimal design, which reduces the length greatly while retaining the high accuracy.
>
> |  | acc(↑) | think len (↓) |
> | --- | --- | --- |
> | GRPO | 37.0 | 272 |
> | DAPO | 48.7 | 296 |
> | TON (ours) | 51.0 | 96 |
>
> [1] “DAPO: An open-source llm reinforcement learning system at scale.” arXiv preprint arXiv:2503.14476 (2025).
>
>
> # **W3: Would be better to apply TON on other or larger models.**
>
> ---
>
> Thanks for your suggestion. We add the experiments on **InternVL3-2B and 8B as follows**. Our TON can retain comparable accuracy while reducing the think length significantly, particularly, it reduces length further on a stronger model (8B), which **demonstrates the generalization of TON across different backbones.**
> Due to time limitations, more results on the bigger VLM, such as Qwen2.5-VL-32B, will be included in our next version.
>
> | Models | Metrics  | Zero-shot | GRPO | TON |
> | --- | --- | --- | --- | --- |
> | InternVL3-2b | acc(↑) | 28.7 | 31.9 | **32.8** |
> |  | think len(↓) | 1204 | 1275 | **364** |
> |  InternVL3-8b | acc(↑) | 45.5 | 65.1 | **65.4** |
> |  | think len(↓) | 750 | 724 | **341** |
>
> ---
>
> We appreciate your questions and comments very much. Please let us know for any further questions.

---

> > ### Author Response · Authors · 2025-08-04
> > **Kind Tips**
> >
> > Dear Reviewer WjY5,
> >
> > Thank you once again for your feedback and hope you have a good rest!
> >
> > Since the rebuttal period is halfway through, we would greatly appreciate it if you could review our response to ensure it adequately addresses your concerns. And we sincerely hope that you can reconsider our work in light of these clarifications. If you have any further comments, please do not hesitate to contact us.
> >
> > We greatly appreciate your selfless contributions to the community. Thank you for your time and consideration.
> >
> > Best regards,
> >
> > Authors of **Think or Not? Selective Reasoning via Reinforcement Learning for Vision-Language Models**

---

> > ### Author Response · Authors · 2025-08-08
> >
> > Dear Reviewer WjY5,
> >
> > I hope this message finds you well. As the discussion phase nears its end, with just **one day remaining**, I would like to ensure that we have thoroughly addressed all your concerns. If there are any additional points of feedback you would like us to consider, please let us know. Your insights are invaluable to us, and we’re eager to address any remaining issues to improve our work.  We sincerely hope that you could reconsider our work in light of these clarifications.
> >
> > Thank you for your time and effort in reviewing our paper.
> >
> > Authors of Submission 15147

---

> > ### Comment · Reviewer_WjY5 · 2025-08-09
> >
> > Thanks for the detailed response. After reading the rebuttal and other reviewers' comments, some of my concerns have been addressed. However, I still think that random dropping could potentially have negative impacts in some cases, and that designing an adaptive method would be a much better approach. Although the empirical results presented in the rebuttal show that the random method performs slightly better than the adaptive method, this could be due to the current adaptive method containing some noisy signals or not being sufficiently strong. Additionally, the empirical results supporting this conclusion are quite limited (only tested on Qwen2.5-VL-3b with GeoQA). Regarding the adaptive reasoning method in LLMs, I think "when to think" is a subset of "how to think." The "no-think" mode here might be similar to the special case of controlling the think strength/length with a zero-think length. Moreover, there are already many existing papers addressing the "how to think" aspect in LLMs.

---

> ### Comment · Area_Chair_p76H · 2025-08-08
>
> Dear Reviewer WjY5,
>
> This is the last call for author-reviewer discussion, which will ends on Aug 8. Could you please read authors' rebuttal and confirm whether your concerns have been addressed asap? Thank you.
>
> Best,
>
> AC

---

> ### Author Response · Authors · 2025-08-09
> **We respectfully disagree with the statements, which we believe are unfair**
>
> Thanks for your reply. We respectfully **disagree** with your statements, which we believe are unfair. Our detailed clarifications for each point are provided below.
>
> > **1. TON uses random thought in SFT, but remains adaptive in GRPO.**
>
> - The SFT stage is **solely** for *format following*—whether to “think” or not is part of that format—as evidenced in Section 5.4 (*Empirical Verification of SFT Significance in TON*). We explicitly **separate format following from reasoning ability**: random dropout in SFT establishes the format, while reasoning ability is developed in the GRPO stage. In GRPO, the model adapts freely, learning when to skip unnecessary steps to optimize RL performance, thus **remaining adaptive**. In other words, `adaptiveness does not need to be introduced in the SFT stage, as each stage targets a distinct objective`'.
>
> - This adaptiveness mirrors standard dropout in neural networks—randomness during training still enhances generalization, as the network learns to handle varied conditions.
>
> > **2. Does human design in SFT really equal an adaptive method?**
>
> - Human design inevitably introduces **heuristics**. In our experiments, the straightforward difficulty-aware design (judging necessity based on correctness) achieved comparable accuracy to—and even slightly lower than—our random dropout method. This suggests that *a carefully crafted SFT strategy has limited influence on final performance, making human heuristics **largely unnecessary***.
>
> - Moreover, creating a single human-defined assumption that generalizes across diverse setups is inherently difficult. In our experiments spanning math, visual, and agent tasks, the more robust and reliable approach is to **let the model explore autonomously within its specific environment during the RL GRPO stage**.
>
> > **3. ‘When to Think’ vs. ‘How to Think’?**
>
> - The reviewer WjY5 claims that `“When to Think” is a subset of “How to Think,”` which we **disagree with**. The distinction between "think" and "non-think" modes is non-trivial. The significance of the “Think-or-not” problem is also recognized by the reviewer vGCC, g1TJ, WjY5, bSRJ. (**vGCC** highlights that "the valuable practice value in real-world applications.",  **g1TJ** notes that "the idea to equip VLMs with selective reasoning is practical"). In Fig. 5d (*G.7 Prompt vs. SFT on different benchmarks*), we tested using only the GRPO stage with hybrid prompts—asking the model to skip thinking if confident—but the skip ratio remained 0. We attribute this to VLMs’ strong conservative bias toward full reasoning, leading to an almost 100% think rate.
>
> - Recent top-performing models such as Gemini-2.5-Flash [1], Qwen3 [2], and GPT-5 [3] underscore the importance of routing between thinking and non-thinking modes. Gemini-2.5-Flash and Qwen3 even provide a `manual` switch to toggle between the two modes, while GPT-5 introduces an automated router that decides based on the user query. **These findings highlight the significance and practicality of “When to Think,” showing it is a distinct, impactful, and unsolved challenge—not merely a subset of “How to Think.”**
>
> [1] Gemini 2.5: Pushing the Frontier with Advanced Reasoning, Multimodality, Long Context, and Next Generation Agentic Capabilities.
>
> [2] Qwen3 Technical Report
>
> [3] GPT-5 System Card
>
> > **4. Many existing papers on “How to Think” in LLMs.**
>
> - Our work focuses on *“When to Think”* rather than *“How to Think”*, as **the first to use vision-language models as the testbed** across diverse setups—math, visual counting, and **agent** tasks. We show the substantial potential of the think-or-not paradigm in VLMs to improve efficiency without sacrificing performance.
>
> - In an orthogonal direction, the newest LLM methods such as AdaCoT, ThinkLess, and LASER [4–6] focus on *how to think*. Most works were released on arXiv only after the NeurIPS full submission deadline and therefore **do not** diminish *the novelty, originality, or timeliness of our contribution*.
>
> [4] *AdaCoT: Pareto-Optimal Adaptive Chain-of-Thought Triggering via Reinforcement Learning*. arXiv:2505.11896 (2025).
> [5] *ThinkLess: LLM Learns When to Think*. arXiv:2505.13379 (2025).
> [6] *Learn to Reason Efficiently with Adaptive Length-Based Reward Shaping*. arXiv:2505.15612 (2025).

---

### Note · Authors · 2025-08-12

We sincerely thank all reviewers and the AC for detailed feedback and constructive engagement. We appreciate the recognition of our work’s novelty, motivation, thorough experiments, and practical value. Following our rebuttal:

* **PYs6** raised their score from 3 → 4.
* **vGCC** maintained a positive score of 4 with all concerns resolved.
* **bSRJ** rated 5, explicitly noting no significant weaknesses remain.

Remaining reservations from **g1TJ** and **WjY5** were already addressed in our rebuttal:

* **g1TJ**’s only concern is that the difficulty-aware dropout ablation performs worse, seemingly misinterpreted as our core method. It was included *solely as an ablation* to show why heuristic-free random dropout is more effective.
* **WjY5** asked (1) Our adaptiveness vs. difficulty-aware designs in SFT, (2) scalability to other models, and (3) distinction from length-adaptive methods.

1. **Highlighted by our title – Adaptiveness via RL** – We emphasize that **our straightforward design stems from extensive analysis**. (A) The difficulty-aware dropout ablation (easy/hard by correct/wrong) performs slightly worse than random dropout, confirming that defining “task difficulty” via heuristics is unnecessary. (B) Sec. 5.4 explain why *format following* should be handled in SFT, while *reasoning ability* is developed in GRPO. (C) Skip-ratio analysis (Sec. A.1) shows TON skips easy tasks yet reasons on harder ones, showing adaptive capacity by GRPO.

2. **Generality and scalability** – TON applies to larger VLMs (InternVL3-2B, 8B) and LLM benchmarks (GSM8K), covering **four models across six setups**. It consistently proves robust across scales and modalities.

3. **Distinct from adaptive reasoning** – Length-adaptive methods assume reasoning is always needed. While *whether to think*, making it orthogonal and complementary:

| Method type | Modalities | Focus             | Setups                      |
| ----------- | ---------- | ----------------- | --------------------------- |
| Existing    | LLM        | Length adaptation | Mainly math                 |
| TON (ours)  | VLM        | Whether to think  | Math, counting, *agent* tasks |

In summary, TON is **the first to explore “Think-or-Not” in VLMs**. Recent top models (Qwen3, GPT-5) reasoning routing designs, underscoring “When to Think” as a distinct challenge.

We believe all concerns have been addressed and respectfully invite reviewers and the AC to revisit them carefully before final recommendations.

---

### Decision · Program_Chairs · 2025-09-17

**Decision:**

Accept (poster)

**Comment:**

This paper studies the issue of long CoT preference for large VLMs, which wastes computation and increases latency. This work further proposed a two-stage training framework combining with SFT and GRPO. The main novel designs are (1) SFT with random thought dropout and (2) selective reasoning reward to avoid unnecessary reasoning. Evaluations are conducted across diverse vision-language tasks under both 3B and 7B models, validating the effectiveness of the proposed method.

The main strengths are (1) the problem and solution to control when to think for VLMs are clear and practical, (2) the empirical gains in both efficiency and accuracy is significant, (3) the paper is well-written and easy to follow. The major weaknesses and concerns are (1) random thought dropout may hurt complex reasoning and the solution should be principled, (2) the comparisons to ThinkLess/LASER/AdaCoT and for text-only evaluations should be clearer, (3) the clarity on GRPO rewards & training strategy needs to be improved, (4) the sensitivity of drop-out rate needs forther ablations.

After rebuttal and discussions, most of the concerns are well addresses. The remaining concerns are that (1) difficulty-aware dropout ablation performs worse, (2) empirical results are limited to Qwen2.5-VL-3b with GeoQA. AC also checked authors' final remarks on the concerns, and believe that these concern are not major reasons for rejecting the paper. For difficulty-aware dropout ablation performs worse, AC agreed with authors that it is not core contribution of this paper but a complimentary ablation to analyze heuristic-free random dropout. For model&dataset selection, this paper also conducted experiments with Qwen2.5-VL-7b and CLEVR. AC didn't find strong evidence to support reviewers' claim on this point. Considering the overall quality of the paper and the concerns are well addressed, AC recommended accepting this paper.